# REFLEXIVE GUIDANCE: IMPROVING OODD IN VISION-LANGUAGE MODELS VIA SELF-GUIDED IMAGE-ADAPTIVE CONCEPT GENERATION

**Jihyo Kim**[*], **Seulbi Lee**[*], **Sangheum Hwang**[†]
Department of Data Science, Seoul National University of Science and Technology
{jihyo.kim, seulbi, shwang}@ds.seoultech.ac.kr

## ABSTRACT

With the recent emergence of foundation models trained on internet-scale data and demonstrating remarkable generalization capabilities, such foundation models have become more widely adopted, leading to an expanding range of application domains. Despite this rapid proliferation, the trustworthiness of foundation models remains underexplored. Specifically, the out-of-distribution detection (OoDD) capabilities of large vision-language models (LVLMs), such as GPT-4o, which are trained on massive multi-modal data, have not been sufficiently addressed. The disparity between their demonstrated potential and practical reliability raises concerns regarding the safe and trustworthy deployment of foundation models. To address this gap, we evaluate and analyze the OoDD capabilities of various proprietary and open-source LVLMs. Our investigation contributes to a better understanding of how these foundation models represent confidence scores through their generated natural language responses. Furthermore, we propose a self-guided prompting approach, termed *Reflexive Guidance (ReGuide)*, aimed at enhancing the OoDD capability of LVLMs by leveraging self-generated image-adaptive concept suggestions. Experimental results demonstrate that our ReGuide enhances the performance of current LVLMs in both image classification and OoDD tasks.[1]

## 1 INTRODUCTION

Thanks to substantial advancements in hardware and the availability of large-scale datasets, foundation models have achieved remarkable performance across a wide range of tasks, demonstrating exceptional generalization. This evolution has shifted toward leveraging multiple data modalities, particularly vision and natural language. As a result, vision-language foundation models have demonstrated their capabilities across diverse domains, from general tasks like text-to-image generation and vision-question answering, to specialized fields such as medical diagnosis

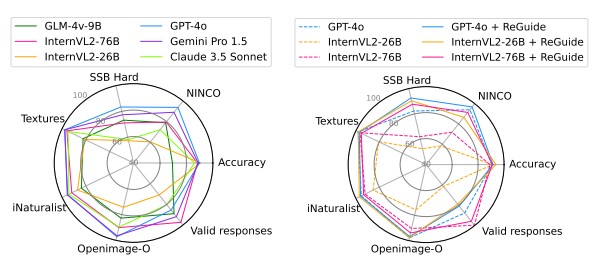

(a) Comparison of LVLMs    (b) Effectiveness of ReGuide

Figure 1: Performance of proprietary and open-source LVLMs on the ImageNet200 benchmark: (a) state-of-the-art comparison and (b) the boosting effect of our ReGuide

and robotics (Esser et al., 2024; Li et al., 2023; Wake et al., 2023; Majumdar et al., 2024). Despite the widespread adoption of these large vision-language models (LVLMs) including GPT (OpenAI, 2024), Claude (Anthropic, 2024), and Gemini (Reid et al., 2024), their trustworthiness and reliability have not been adequately investigated. Ensuring the robustness of deep neural networks has been a key research area to guarantee their safe application in practice. With the rapid popularization of LVLMs, practical concerns such as harmful content filtering and domain generalization have

---

[*]Equal contribution
[†]Corresponding author
[1]https://github.com/daintlab/ReGuide

been actively studied (Zhang et al., 2024b; Han et al., 2024). However, fundamental aspects, such as the quality of confidence estimates for model predictions—extensively studied in single-modal models—remain underexplored. Some works have studied OoDD in CLIP (Ming et al., 2022; Jiang et al., 2024; Cao et al., 2024) and Stable Diffusion (Zhu et al., 2024), but few have examined OoDD in LVLMs, particularly with respect to confidence scores expressed through their natural language responses. This gap between their demonstrated capabilities and real-world reliability raises concerns about ensuring the safe and dependable deployment of LVLMs.

To bridge the gap, we first evaluate and compare the OoDD capabilities of LVLMs. The OoDD task allows us to investigate how LVLMs behave when required to generate responses beyond the categories defined within the user-provided prompt. Due to the lack of prior experimental configurations, we develop a framework for evaluating the OoDD capabilities of LVLMs. Our focus is on detecting OoD image inputs based on the in-distribution (ID) space specified by text inputs. To this end, we design a prompt that defines the ID space and guides the LVLM to provide confidence estimates through its generated responses. Using this prompt, we evaluate the OoDD capabilities of both proprietary and open-source LVLMs and analyze their behavior in expressing confidence from various perspectives. Fig. 1(a) summarizes the comparison of the evaluated LVLMs. In general, proprietary models outperform open-source models, and open-source models show performance improvements as model size increases. However, some open-source models including LLaVa-v1.6 (Mistral-7B) (Li et al., 2024a) and GLM-4v-9B (GLM et al., 2024) exhibit low OoDD performance despite achieving decent results on popular VLM benchmarks.[2] We found that one of the reasons for this discrepancy is the insufficient image interpretation capabilities of these models, which limits their ability to accurately distinguish between ID and OoD inputs.

To enhance the OoD detectability of LVLMs, we propose a two-stage self-guided prompting approach called *Reflexive Guidance (ReGuide)*. In the first stage, ReGuide prompts an LVLM to suggest two groups of concepts based on the given image: semantically similar concepts (i.e., near-OoD) and semantically dissimilar concepts (i.e., far-OoD). In the second stage, these suggested concepts are employed as auxiliary OoD classes. To the best of our knowledge, this is the first study to leverage image inputs to generate informative texts for OoDD. By utilizing the visual interpretation capabilities of LVLMs, ReGuide remains simple and model-agnostic, allowing the same prompt to be applied to different LVLMs. Fig. 1(b) shows the effectiveness of ReGuide. Notably, ReGuide significantly boosts the overall performance of open-source models, making them comparable to proprietary models. GPT-4o, which is the top performer prior to applying ReGuide, also benefits from ReGuide, especially on near-OoD datasets (*e.g.*, NINCO, SSB Hard). Our results highlight the effectiveness of guiding LVLMs through self-generated, image-adaptive concept suggestions.

From the results of our study, we can draw the following insights regarding the effectiveness of ReGuide: Despite the strong visual interpretation capabilities of LVLMs, which enable them to predict fine-grained classes of objects effectively, these models tend to avoid generating responses that fall outside the given prompt categories. Our findings suggest that the models may have developed a form of positive bias due to their training on positive image-text pairs, or that their ability to explore areas beyond the information embedded in the prompt has been diminished, possibly due to human alignment processes. Addressing these limitations can lead to more reliable and versatile applications of LVLMs in various domains.

## 2 RELATED WORK

In single-modal vision models, OoDD has been actively studied. Starting with the baseline method of using the maximum softmax value as an OoD score (Hendrycks & Gimpel, 2017), methods have evolved to improve confidence modeling (Moon et al., 2020; Liu et al., 2020; Bibas et al., 2021), post-hoc techniques (Liang et al., 2018; Lee et al., 2018b; Sun et al., 2021; Djurisic et al., 2023; Sun et al., 2022; Zhang et al., 2023; Liu & Qin, 2024), or a combination of both (Xu et al., 2024). Another approach involves leveraging auxiliary OoD samples to better distinguish between ID and OoD inputs. While real OoD samples show strong detection performance (Hendrycks et al., 2019; Chen et al., 2021; Lu et al., 2023; Katz-Samuels et al., 2022), they require access to real OoD data. Synthetic samples offer an alternative, providing the benefits of auxiliary OoD data without the need for collecting real samples (Lee et al., 2018a; Du et al., 2022; Tao et al., 2023; Zheng et al., 2023).

---

[2] https://huggingface.co/spaces/opencompass/open_vlm_leaderboard

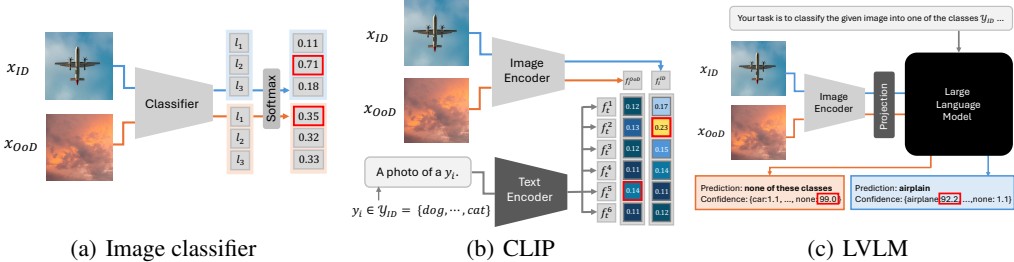

(a) Image classifier      (b) CLIP      (c) LVLM

Figure 2: Comparison of the OoDD framework for single-modal classifiers, CLIP, and LVLMs

With the growing interest in multi-modal models, particularly VLMs, OoDD methods have advanced toward leveraging both vision and language modalities. A pioneering LVLM (non-generative) is CLIP (Radford et al., 2021). Ming et al. (2022) established a baseline for OoDD on CLIP by using the maximum cosine similarity between image and text features as an OoD score. One main approach in CLIP-based OoDD focuses on using auxiliary text for OoD concepts, as text data is relatively easy to collect. Wang et al. (2023) simply added 'no' as a prefix to ID classes, while Fort et al. (2021) used labels from OoD datasets, which requires prior knowledge of OoD inputs. Jiang et al. (2024); Esmaeilpour et al. (2022); Cao et al. (2024) derived auxiliary texts using only ID data, with Cao et al. (2024) obtaining OoD texts from large language models (LLMs) like GPT. Previous work such as NegLabel (Jiang et al., 2024) and EOE (Cao et al., 2024), as well as our proposed method, share the idea of using negative concepts for OoDD. However, we focus on generative LVLMs and our approach is the first to utilize image-adaptive negative concepts generated by LVLMs.

Diffusion models, a type of generative model, have also been employed for OoDD. One approach is to compare the reconstruction quality between ID and OoD images (Gao et al., 2023; Tong & Dai, 2024), or to generate synthetic images for OoDD in single-modal vision models (Du et al., 2023; Girella et al., 2024). However, generative VLMs, particularly foundation models (i.e., LVLMs), remain understudied in the context of OoDD. While several works have explored confidence estimates in model predictions (Han et al., 2024; Groot & Valdenegro-Toro, 2024), they do not focus on detecting OoD inputs. Miyai et al. (2024) introduced Unsolvable Problem Detection to evaluate VLMs' ability to reject unanswerable inputs. However, their focus is on rejection ability, while our study focuses on distinguishing between answerable and unanswerable inputs and proposes a method to enhance this ability. In LLMs, several works use model-generated answers to guide final responses more accurately (Yao et al., 2023; Wei et al., 2022; Shanahan et al., 2023). Our work is the first to propose a similar approach in LVLMs, leveraging the model's own generated answers.

## 3 OoD Detection on Vision-language Foundation Models

### 3.1 Problem Definition

OoD is conventionally defined as distributions outside the training distribution. However, given the vast amount and broad domain coverage of data used to train LVLMs, this conventional definition faces challenges in its direct application to LVLMs. To address this, we extend the zero-shot OoDD framework of CLIP (Radford et al., 2021) to generative LVLMs.

Let $\mathcal{X}$ be the image space and $\mathcal{Y} = \{y_i\}_{i=1}^{C}$ the set of class-representing words, where $C$ is the number of classes. The OoDD problem for CLIP in zero-shot image recognition is defined as the scenario where $\mathcal{Y}$ does not contain the ground-truth label of an input image $\mathbf{x} \in \mathcal{X}$. Given $\mathbf{x}$, CLIP yields a prediction for $\mathbf{x}$ based on $sim(f_I(\mathbf{x}), f_T(\texttt{prompt}(y_i)))$ where $sim(u, v)$ is the cosine similarity, $\texttt{prompt}$ is a text template designed to reflect $y_i$ (e.g., a photo of $y_i$), and $f_I$, $f_T$ are the image and text encoders, respectively. The cosine similarities between $\mathbf{x}$ and each $y_i \in \mathcal{Y}$ are used to determine whether $\mathbf{x}$ belongs to ID or OoD. If the ground-truth label of $\mathbf{x}$ is not in $\mathcal{Y}$, those similarities for all $y_i$ in $\mathcal{Y}$ will be relatively low, leading to the classification of $\mathbf{x}$ as OoD.

We frame the OoDD problem for LVLMs based on the zero-shot OoDD scenario defined for CLIP. Similarly, $\mathcal{Y}$ is set to a fixed word set containing only ID class words. Given both $\mathbf{x}$ and $\mathcal{Y}$ as inputs, an LVLM is instructed to produce prediction results in a structured format. Then, class predictions and confidence estimates for $\mathbf{x}$ are extracted from the natural language responses by the LVLM. If the ground-truth label of $\mathbf{x}$ is not in $\mathcal{Y}$, the LVLM should provide low confidence scores for all $y_i$ in

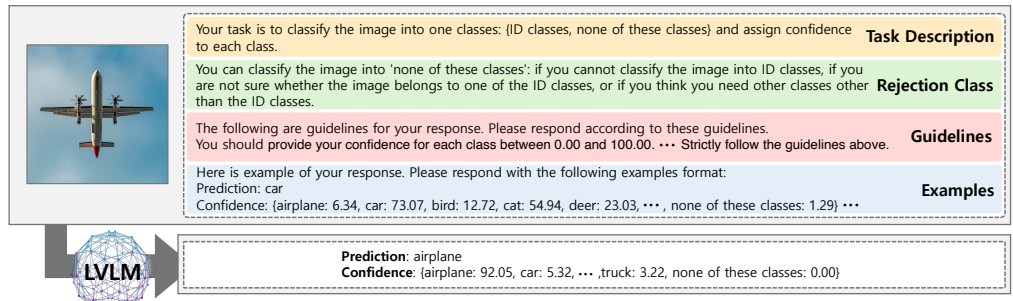

Figure 3: Framework for OoDD evaluation on LVLMs

$\mathcal{Y}$ in its answers. Fig. 2 summarizes the OoDD frameworks for single-modal image classifiers, CLIP, and LVLMs. Single-modal classifiers detect OoD images based solely on outputs from image inputs. In contrast, CLIP and LVLMs leverage both image and text inputs, with CLIP combining them via similarity measures and LVLMs integrating them through cross-modal interactions. LVLMs also generate natural language outputs, controllable through prompt instructions.

## 3.2 PROMPT DESIGN

Fig. 3 illustrates the framework for OoDD evaluation on LVLMs with a simplified prompt. Our prompt consists of four components: a task description, an explanation of the rejection class, guidelines, and examples for the response format. Unlike previous work on LVLMs (Han et al., 2024; Hwang et al., 2024; Groot & Valdenegro-Toro, 2024), we encounter a significant number of failure cases when using a simple prompt consisting of the task statement including $\mathcal{Y}$ and a formatted output structure. We attribute this inconsistency to differences in how confidence scores are assigned in our framework. In our prompt design, the LVLM is expected to provide confidence scores for each class in $\mathcal{Y}$, whereas prior work required only a single confidence score for the predicted class. To mitigate these failures, we enhance the prompt by adding the following components, resulting in the final prompt used in our experiments. The effectiveness of providing guidelines and examples and the complete prompt can be found in Appendix B.5 and B.6, respectively.

**Rejection class.** When we provide only $\mathcal{Y}$, we observe that predictions for OoD samples often do not correspond to any class in $\mathcal{Y}$. Due to its strong zero-shot visual recognition capabilities, the LVLM can either provide the ground-truth label for an OoD input image or reject classification into any of the classes in $\mathcal{Y}$. To address this unintended behavior, we introduce a rejection class named *none of these classes* and provide a detailed explanation of its role to the LVLM.

**Guidelines addressing failure cases.** In addition to the case where the prediction does not fall within $\mathcal{Y}$, we identify several other issues, such as mismatches between the class assigned the highest confidence score and the predicted class, assigning a confidence score of 0.0 to all classes including the rejection class, or failing to provide confidence scores for all classes in $\mathcal{Y}$. To reduce the frequency of these failures, we include guidelines that address the most common cases.

**Response examples.** In-context learning (ICL) is commonly used to improve response quality by providing models with examples that illustrate the desired format during inference, allowing them to generate outputs that better align with these examples (Brown et al., 2020). We leverage ICL to reduce failure caused by formatting issues. However, we observe that when only a single example is provided, the LVLM tends to mimic the example, regardless of whether the input image is ID or OoD. To mitigate this tendency, we provide examples for both ID and OoD input images.

## 3.3 OoD SCORE DESIGN

Since we provide the rejection class for OoD inputs, the ideal behavior of LVLMs for confidence estimates is to assign high confidence scores to one of the classes in $\mathcal{Y}$ for ID inputs, and to the rejection class (i.e., *none of these classes*) for OoD inputs. Therefore, we use the maximum confidence score among the classes in $\mathcal{Y}$ (i.e., ID classes) as the OoD score. Note that we do not constrain the sum of confidence scores. We apply the softmax function to all confidence values to normalize them, including that of the rejection class. Based on this OoD score design, an input image is likely to be ID if the score is high, and likely to be OoD if the score is low.

Table 1: Comparison on the ImageNet200 benchmark. Full model names are in the footnotes. 'Valid' indicates the ratio of valid responses out of a total of 23,031 image-prompt queries, with counts in brackets. **Bold** highlights the best performance among generative LVLMs.

| Models | ID IN200 | | Near-OoD | | Far-OoD | | | All OoD |
|---|---|---|---|---|---|---|---|---|
| | | | NINCO | SSB-Hard | iNaturalist | Textures | Openimage-O | |
| | Valid | ACC (↑) | FPR@95%TPR (↓) / AUROC (↑) | | | | | |
| SCALE** | - | 86.37 | 84.84 | | 93.98 | | | - |
| fDBD** | - | 86.37 | 84.27 | | 93.45 | | | - |
| AugMix+ASH** | - | 87.01 | 55.83 / 85.74 | 71.22 / 80.00 | 19.14 / 95.81 | 21.00 / 95.67 | 31.06 / 92.51 | - |
| OpenCLIP | 100.00 (23,031) | 87.41 | 62.27 / 85.31 | 71.48 / 78.36 | 42.76 / 92.49 | 47.83 / 89.62 | 47.47 / 90.68 | 61.42 / 83.54 |
| GPT-4o | 85.49 (19,689) | 89.78 | 22.30 / **92.08** | **38.95 / 81.41** | 2.06 / 97.58 | 7.45 / **95.85** | **3.78 / 97.17** | **23.50 / 88.50** |
| Claude 3.5 Sonnet | 80.39 (18,515) | 86.06 | 52.92 / 72.18 | 78.41 / 58.09 | 9.23 / 94.93 | 10.17 / 94.28 | 18.31 / 89.71 | 49.01 / 73.69 |
| Gemini Pro 1.5 | 91.92 (21,170) | 88.84 | **21.55** / 89.03 | 55.24 / 77.40 | **1.53 / 97.73** | **5.12** / 95.61 | 5.25 / 96.45 | 32.97 / 85.74 |
| LLaVA-v1.6 | 71.63 (16,496) | 2.45 | 100.00 / 50.85 | 100.00 / 48.95 | 100.00 / 50.05 | 100.00 / 59.26 | 100.00 / 49.23 | 100.00 / 50.11 |
| GLM-4v | 89.00 (20,498) | 69.41 | 100.00 / 79.23 | 100.00 / 74.35 | 100.00 / 83.01 | 100.00 / 83.45 | 100.00 / 83.11 | 100.00 / 77.86 |
| InternVL2-26B | 62.68 (14,436) | **90.22** | 82.59 / 58.32 | 94.21 / 52.51 | 36.69 / 81.26 | 28.08 / 85.56 | 50.89 / 74.16 | 75.95 / 61.63 |
| InternVL2-76B | 97.36 (22,424) | 88.30 | 100.00 / 72.27 | 100.00 / 62.39 | 100.00 / 95.57 | 100.00 / 91.62 | 100.00 / 90.12 | 100.00 / 74.14 |

[*] OpenCLIP-ViT-B-32, GPT-4o (2024-08-06), LLaVA-v1.6-Mistral-7B, GLM-4v-9B, InternVL2-InternLM2-Chat-26B, InternVL2-LLaMA3-76B
[**] Results based on 100% of the benchmark from the OpenOOD v1.5 leaderboard.[3] Only the results available from the leaderboard are shown.

## 3.4 EXPERIMENTAL SETTINGS

**Comparison models.** To compare LVLMs from diverse perspectives, we consider both proprietary and open-source models. For proprietary models, we employ three state-of-the-art models: GPT-4o (2024-08-06) (OpenAI, 2024), Gemini Pro 1.5 (Reid et al., 2024), and Claude 3.5 Sonnet (Anthropic, 2024). For open-source models, we use four models: LLaVA-v1.6-Mistral-7B (LLaVA-v1.6) (Li et al., 2024a), GLM-4v-9B (GLM-4v) (GLM et al., 2024), QWEN-VL-Chat (QWEN) (Bai et al., 2023), InternVL2-InternLM2-Chat-26B (InternVL2-26B), and InternVL2-LLaMA3-76B (InternVL2-76B) (Chen et al., 2024). Additionally, we include OpenCLIP (ViT-B-32 pretrained on LAION 2B-s34b-b79k) (Cherti et al., 2023) as a non-generative LVLM. To further facilitate comparison between single- and multi-modal models, we also include three single-modal SOTA vision OoDD models, SCALE (Xu et al., 2024), fDBD (Liu & Qin, 2024), Aug-Mix+ASH (Djurisic et al., 2023; Hendrycks et al., 2020).

**Benchmark datasets.** We evaluate the comparison models on the CIFAR10 and ImageNet200 benchmarks proposed in OpenOOD v1.5 (Zhang et al., 2024a). We focus on the standard OoD setting in OpenOOD v1.5, which includes two types of datasets, near- and far-OoD, categorized based on the semantic distance between ID and OoD datasets. Since LVLMs are trained on high-resolution images, our main experiments are conducted on the ImageNet200 benchmark. We also evaluate the models on the CIFAR10 benchmark to assess their scalability with respect to input image resolution. For each benchmark, we consider the set of class names from the ID dataset as $\mathcal{Y}$. Due to cost, time, and API rate limits, we use 25% subsets of the benchmarks. A detailed explanation of the benchmark datasets can be found in Appendix B.1.

**Evaluation metrics.** We measure the OoDD performance using two commonly used metrics in OoDD: the area under the receiver operating characteristic curve (AUROC) and the false positive rate at the 95% true positive rate (FPR@95%TPR; FPR). We also include the ratio and number of valid responses—those that adhere to the provided guidelines and example format—as part of the evaluation metrics. We consider the ability to understand given instructions to be one of the model's key capabilities. Only valid responses are included when measuring performance. We additionally evaluate models on the shared valid responses across all models for a rigorous comparison. This result can be found in Appendix B.2.

## 3.5 RESULTS

Tab. 1 presents the OoDD capabilities of the compared models on the ImageNet200 benchmark. The near- and far-OoD results for SCALE and fDBD represent the average performance across their respective categories. 'All OoD' refers to the performance in distinguishing all OoD inputs, including near- and far-OoD inputs, from ID inputs (i.e., ID vs. all OoD). The results of QWEN-VL-Chat are ommitted due to its exceptionally low ability to follow instructions, with a valid response rate of less than 1%.

---

[3] https://zjysteven.github.io/OpenOOD/index.html

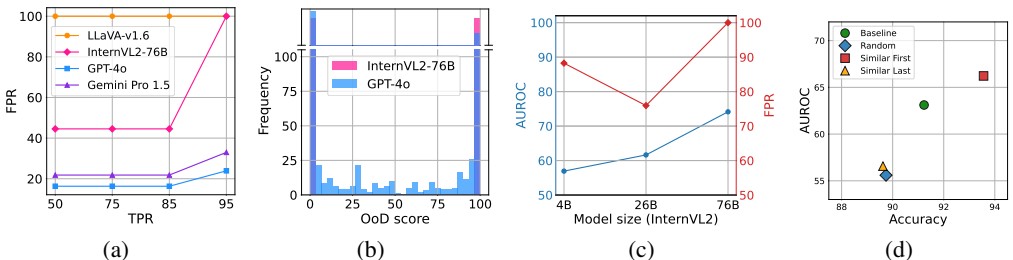

Figure 4: Further analysis on the ImageNet200 benchmark. (a) FPR across different TPR thresholds, (b) OoD score distribution, (c) OoD detectability based on model sizes, and (d) ID classification accuracy and OoDD performance of InternVL2-26B according to the given class order

On both image recognition and OoDD tasks, proprietary models outperform the open-source models in most cases, with reasonable valid response rates. All compared models have more difficulty in detecting near-OoD than far-OoD. The overall performance ranking of the evaluated LVLMs generally aligns with the OpenVLM leaderboard[4], except for Gemini Pro 1.5. In our results, Gemini Pro 1.5 shows better ID accuracy and OoDD performance than Claude 3.5 Sonnet and InternVL-76B. Claude 3.5 Sonnet frequently generates invalid responses compared to other proprietary models and struggles to detect near-OoD, resulting in worse performance on NINCO and SSB-Hard compared to the open-source model GLM-4v in terms of AUROC, despite achieving $16.65\%$ higher ID accuracy. InternVL2-26B achieves the best ID accuracy, but has difficulty with OoDD and shows the lowest valid response rate. While there is no clear-cut relationship between image recognition and OoDD capabilities, models with better image recognition performance generally exhibit stronger OoDD performance, consistent with the findings of Vaze et al. (2022).

Notably, the proprietary models generally perform on par with or better than the single-modal SOTA OoDD models. In addition, GPT-4o and Gemini Pro 1.5 outperform OpenCLIP in both ID classification and OoDD, showing significantly better performance in OoDD. For a more rigorous comparison, we evaluate OpenCLIP on images where GPT-4o generates valid responses, as GPT-4o has a relatively lower valid response rate. On the GPT-4o valid query set, GPT-4o still yields better results on both tasks. This demonstrates its superior visual recognition capability and ability to express confidence scores through its generated responses. This observation is consistent with the evaluation on the shared valid set across all models compared. Detailed results can be found in Appendix B.2.

Among the open-source models, one notable observation is their FPR of $100\%$. Our analysis indicates that this is caused by extremely biased confidence values, which will be further discussed in Sec. 3.6. GLM-4v exhibits the best OoDD performance, while InternVL2-76B shows the second-best zero-shot image recognition performance among the open-source models. Both models also demonstrate a better understanding of instructions than Claude 3.5 Sonnet. Overall, InternVL2-76B shows superior performance in all aspects, including valid response rate, ID accuracy, and OoDD performance. It achieves higher ID accuracy and comparable OoDD performance to the single-modal SOTA models, alongside the highest valid response rates among all compared models. LLaVA-v1.6 struggles with image recognition, achieving an accuracy of just $2.45\%$. Most responses from LLaVA-v1.6 incorrectly identify ID images as OoD, resulting in poor classification accuracy and OoDD performance.

## 3.6 FURTHER ANALYSIS

**Scalability with image resolution.** We assess the input resolution scalability of LVLMs using the CIFAR10 benchmark. Most models demonstrate higher performance on the CIFAR10 benchmark than on the ImageNet200 benchmark, with a higher ratio of valid responses. Although the smaller number of class candidates and coarser class contribute to the improved performance, it indicates that the LVLMs handle lower-resolution images effectively. As with the ImageNet200 benchmark, proprietary models consistently outperform open-source models across all datasets, demonstrating superior performance even on low-resolution images. However, in terms of valid response rate, only GPT-4o shows a decrease, excluding GLM-4v which is significantly less accurate overall. This suggests that GPT-4o has more difficulty handling low-resolution images compared to high-resolution images. Detailed results can be found in Appendix B.2.

---

[4] https://huggingface.co/spaces/opencompass/open_vlm_leaderboard

Table 2: Reasoning behind the predictions of GPT-4o and InternVL2-26B on the ImageNet200 benchmark. **Bold** indicates the predicted class and its confidence score before applying softmax. ***Bold italic*** highlights notable points. Classes with a confidence score of 0.0 are omitted.

| | | Prediction | | |
|---|---|---|---|---|
| | | **ID** | | **OoD** |
| **Actual** **ID** **GPT-4o** |  | • Ground-truth: Bald eagle
• {**bald eagle**: **98.75**, vulture: 1.0, none of these classes: 0.25}
• The bird in the image is *a clear match to the bald eagle class*, recognizable by its distinctive **white head, yellow beak, and large wingspan**. Hence, a high confidence is assigned to this class. |  | • Ground-truth: Centipede
• {centipede: 40.00, snail: 45.00, **none of these classes**: **80.00**}
• The image appears to *resemble a millipede-like structure*, but none of the given 200 classes directly match this observation. Thus, the prediction is 'none of these classes' with high confidence. |
| **InternVL2-26B** |  | • Ground-truth: Tree frog
• {**tree frog**: **99.99**}
• The image shows *a green frog with a yellow eye, which is characteristic of a tree frog*. *The frog is perched on a leaf*, and *its coloration and eye pattern are consistent with tree frogs*. |  | • Ground-truth: Bucket
• {**none of these classes**: **100.00**}
• The image shows *a person in a swimsuit, bending over a bucket*, which appears to be in *a grassy area*. The image does not clearly depict any of the 200 classes provided, so the prediction is 'none of these classes' with a confidence of 100.00. |
| **OoD** **GPT-4o** |  | • Ground-truth: N/A (Openimage-O)
• {goldfinch: 5.67, junco: 2.93, bald eagle: 1.42, **lorikeet**: **94.72**, toucan: 8.65, …, snail: 0.09, golden retriever: 0.14, ambulance: 0.18, space shuttle: 0.14, none of these classes: 0.13}
• The bird in the image closely resembles *a lorikeet, with its vibrant green plumage and colorful facial markings*. Other birds like *toucans and goldfinches also show some resemblance*, but *the lorikeet class has the highest confidence due to the specific combination of colors and features*. |  | • Ground-truth: N/A (Openimage-O)
• {**none of these classes**: **99.99**}
• The image shows *purple ornamental flowers (Allium)*, which do not match any of the given 200 classes. |
| **InternVL2-26B** |  | • Ground-truth: Araneus gemma (NINCO)
• {**spider web**: **99.99**}
• The image shows *a spider on a green plant*, with *a web* visible in the background. The *spider's body and legs are clearly visible*, and the *web's intricate pattern is evident*. The confidence in the prediction of 'spider web' is high *due to the clear visibility of the web and the spider's position within it*. |  | • Ground-truth: Caracal (NINCO)
• {**none of these classes**: **99.00**}
• The animal in the image is a *caracal*, which is *a medium-sized wild cat native to Africa and parts of the Middle East. It is characterized by its reddishbrown coat, long legs, and distinctive blacktufted ears. The caracal is known for its agility and is often found in grasslands and savannas*. The image shows an animal that matches these characteristics, making it a "none of these classes" with high confidence. |

**Highly biased confidence scores.** One common observation across both benchmarks is an FPR of 100%. As shown in Fig. 4(a), LLaVA-v1.6 consistently yields 100% FPR across different TPR thresholds. This phenomenon is observed in most open-source models, but not in proprietary models. To understand the cause of this issue, we examine the distribution of OoD scores. Fig. 4(b) depicts the OoD score distribution of InternVL2-76B and GPT-4o on the ImageNet200 benchmark. We find that the compared models tend to produce highly biased OoD scores, with most OoD scores being either 0.0 or 100.00. A detailed explanation can be found in Appendix B.2. This highlights the need for calibration of confidence scores in the responses of open-source models.

**Reasoning.** We analyze the contribution of the model's visual feature interpretability to its OoDD capability by examining the rationale behind its predictions. The prompt used for reasoning can be found in Appendix B.6. Tab. 2 presents the reasoning results from GPT-4o and InternVL2-26B for four different cases on the ImageNet200 benchmark. The results indicate that the high interpretability of visual features contributes to stronger OoDD capabilities. GPT-4o provides detailed descriptions of images, leading to predictions in fine-grained categories. InternVL2-26B also describes objects in a given image effectively, but not with the same level of detail as GPT-4o. A detailed explanation including the reasoning results for LLaVA-v1.6 can be found in Appendix B.3.

**Confidence scores on ID.** We assess confidence scores for ID to rigorously explore the expressiveness of LVLMs in generating confidence estimates. To measure the quality of confidence scores, we employ ECE (Pakdaman Naeini et al., 2015) and AURC (Geifman et al., 2019). As shown in Tab. 3, LVLMs exhibit relatively low ECE despite their highly biased confidence scores. GPT-4o and OpenCLIP demonstrate good quality in predictive confidence for ID, showing lower ECE and AURC compared to other models. Although InternVL2-76B achieves classification accuracy comparable to GPT-4o, its ECE and AURC are significantly worse, indicating that InternVL2-76B assigns high confidence scores to misclassified inputs.

Table 3: Comparison on the quality of confidence scores in ID inputs. **Bold** indicates the best performance excluding OpenCLIP.

| Models | ImageNet200 | | CIFAR10 | |
|---|---|---|---|---|
| | ECE | AURC | ECE | AURC |
| **OpenCLIP** | 2.39 | 21.95 | 2.39 | 11.82 |
| **GPT-4o** | **2.42** | **18.64** | 1.18 | **10.30** |
| **LLaVA-v1.6** | 4.40 | 887.86 | 5.58 | 48.03 |
| **GLM-4** | 3.24 | 87.57 | **0.59** | 283.69 |
| **QWEN** | - | - | 76.82 | 703.73 |
| **InternVL2-76B** | 5.52 | 75.25 | 2.36 | 37.34 |

\* ECE and AURC values are multiplied by $10^2$ and $10^3$, respectively. Lower values are better.

**Scaling law in terms of model size.** We examine the relationship between OoDD capabilities and model size in terms of the number of parameters. Fig. 4(c) shows AUROC and FPR for InternVL2-4B, 26B, and 76B in distinguishing all OoD from ID on the ImageNet200 benchmark. We observe

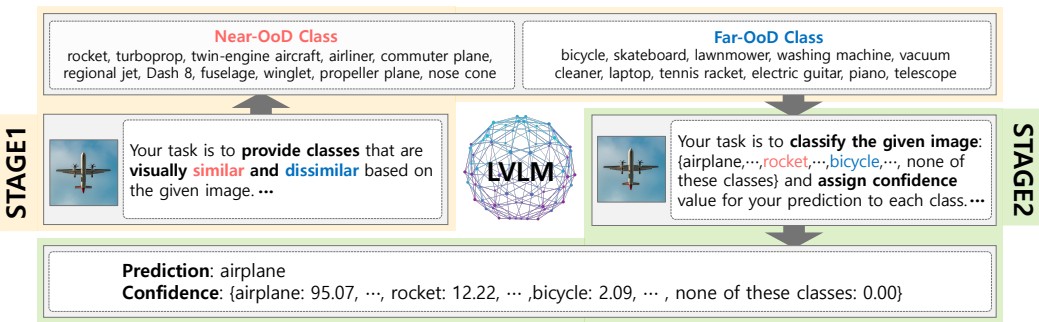

Figure 5: Framework of the proposed Reflexive Guidance for OoDD

that OoD detectability improves as model size increases. This observation aligns with our results and the OpenVLM leaderboard, where larger models generally demonstrate superior performance.

**Class order in the prompt.** To examine the influence of input class order in the prompt, we evaluate InternVL2-26B with different class orders. Detailed experimental settings for class order can be found in Appendix B.4. Fig. 4(d) shows the ID classification accuracy and AUROC (ID vs. all OoD). "Random", "Similar First", and "Similar Last" refer to random ordering, placing similar classes to input images first, and placing similar classes last, respectively. "Similar First" improves both accuracy and OoDD performance, while "Random" and "Similar Last" result in a decrease compared to the baseline. Based on these results, we can infer that the model tends to prioritize inputs received earlier over those received later. We assume that this tendency arises from the structure of the data on which InternVL2-26B is trained. It is observed that in the text of the image-text pair dataset used for the InternVL2-26B training, words referring to objects in the image are predominantly positioned at the beginning of the text (Li et al., 2024c; Wang et al., 2024; Shahroudy et al., 2016). This may have given InternVL2-26B an implicit bias regarding word order in the given prompts.

**Response failure.** Despite providing guidelines to mitigate failures, we still observe failures across all models. A common issue is the mismatch between the predicted class and the class with the highest confidence score. For most proprietary models, the inability to identify the given image is the most frequent failure case. This suggests that proprietary models are more conservative, often declining to provide answers when outcomes are uncertain. One contributing factor to this is their refusal to respond to harmful or inappropriate inputs. All three proprietary models decline to answer questions about these inputs, while none of the open-source models do. Thus, this characteristic of proprietary models may contribute to their low valid response rates. Failures are not confined to any specific dataset, indicating that they stem from both datasets and the underlying mechanisms of models. Detailed results for failure cases and reject responses can be found in Appendix B.5.

## 4 REFLEXIVE GUIDANCE

We introduce a simple and model-agnostic prompting strategy, *Reflexive Guidance (ReGuide)*, to enhance the OoD detectability of LVLMs. The LVLM's strong generalization ability has been demonstrated through its performance across various downstream tasks. Therefore, we leverage the LVLM itself to obtain guidance for OoDD from its powerful zero-shot visual recognition capabilities. Fig. 5 illustrates the overall framework of ReGuide, which is implemented in a two-stage process. Details on the prompts for each stage can be found in Appendix C.5.

**Stage 1: Image-adaptive class suggestions.** In the first stage, the LVLM is asked to suggest $2N$ class names $\mathcal{A}_{\text{aux}}$ derived from the given image. Specifically, we request two groups of class names: 1) $N$ classes that are visually similar to the image denoted as $\mathcal{A}_{\text{aux}}^{\text{near}}$, and 2) $N$ classes that are visually dissimilar or belong to different domains denoted as $\mathcal{A}_{\text{aux}}^{\text{far}}$. In the context of ID, $\mathcal{A}_{\text{aux}}^{\text{near}}$ and $\mathcal{A}_{\text{aux}}^{\text{far}}$ can provide classes conceptually corresponding to near-OoD and far-OoD, respectively. If the input image is OoD, $\mathcal{A}_{\text{aux}} = \mathcal{A}_{\text{aux}}^{\text{near}} \cup \mathcal{A}_{\text{aux}}^{\text{far}}$ can offer potential ground-truth label or class names closely related to the ground-truth label.

**Stage 2: OoDD with suggested classes.** Stage 2 follows a similar procedure to the original OoDD evaluation presented in Fig. 3. The major difference is that $\mathcal{A}_{\text{aux}}$ is employed as auxiliary OoD classes. NegLabel (Jiang et al., 2024) and EOE (Cao et al., 2024) also use auxiliary OoD classes for improving OoDD performance, but they rely on texts to obtain negative concepts. In contrast, our

Table 4: ReGuide effects on the ImageNet200 benchmark. 'Valid' indicates the valid response ratio from 4, 170 queries. **Bold** highlights the best performance among the results from each LVLM.

| Models | ID IN200 Valid | ACC (↑) | Near-OoD NINCO | Near-OoD SSB-Hard | Far-OoD iNaturalist | Far-OoD Textures | Far-OoD Openimage-O | All OoD |
|---|---|---|---|---|---|---|---|---|
| | | | FPR@90%TPR (↓) / FPR@95%TPR (↓) / AUROC (↑) | | | | | |
| InternVL2-26B | 61.01 (2,544) | 91.23 | 82.73 / 82.73 / 58.31 | 94.34 / 94.34 / 52.51 | 38.03 / 38.03 / 80.66 | 28.86 / 28.86 / 85.25 | 47.91 / 47.91 / 75.72 | 73.12 / 73.12 / 63.11 |
| + GPT-text | 69.88 (2,914) | 89.58 | 69.44 / 69.44 / 62.17 | 85.65 / 85.73 / 53.55 | 26.82 / 28.00 / 84.72 | 29.20 / 29.20 / 83.51 | 39.39 / 39.39 / 78.10 | 62.41 / 62.64 / 65.88 |
| + ReGuide | 86.14 (3,592) | 93.53 | 22.39 / 22.89 / 86.53 | 15.21 / 15.21 / 90.41 | 1.39 / 1.39 / 98.02 | 3.93 / 3.93 / 97.05 | 2.04 / 2.04 / 97.68 | 10.24 / 10.27 / 93.19 |
| InternVL2-76B | 97.26 (4,056) | 89.09 | 51.28 / 51.28 / 71.89 | 71.02 / 71.02 / 62.02 | 2.20 / 2.20 / 96.43 | 10.76 / 10.76 / 92.15 | 14.27 / 14.27 / 90.40 | 44.46 / 44.46 / 75.30 |
| +ReGuide | 95.80 (3,995) | 90.93 | 8.05 / 56.36 / 91.35 | 14.58 / 66.65 / 87.65 | 0.00 / 59.75 / 95.35 | 4.08 / 60.00 / 93.38 | 2.02 / 65.46 / 93.95 | 8.92 / 64.36 / 90.60 |
| GPT-4o | 87.58 (3,652) | 90.64 | 8.57 / 14.76 / 93.96 | 29.25 / 34.50 / 82.28 | 0.81 / 1.83 / 98.11 | 5.60 / 6.47 / 95.37 | 1.21 / 3.63 / 97.82 | 15.62 / 19.34 / 89.85 |
| + ReGuide | 79.57 (3,318) | 91.59 | 0.49 / 18.72 / 96.76 | 7.53 / 31.17 / 92.56 | 0.00 / 17.05 / 97.08 | 1.32 / 26.43 / 95.96 | 0.15 / 19.66 / 96.82 | 4.02 / 25.66 / 94.61 |

* GPT-4o (2024-08-06), InternVL2-InternLM2-Chat-26B, InternVL2-LLaMA3-76B

approach utilizes the LVLM, allowing images to be utilized to obtain negative concepts for auxiliary OoD classes. Given the strong zero-shot visual recognition capabilities of LVLMs, it is expected that OoD input images can be assigned higher confidence scores for $\mathcal{A}_{aux}$ than for $\mathcal{Y}$, since $\mathcal{A}_{aux}$ is derived from the input image itself. The rejection class *none of these classes* is retained as a fallback in case $\mathcal{A}_{aux}$ does not adequately function as auxiliary OoD classes.

**OoD score design.** To evaluate the OoDD performance of ReGuide, we employ the same OoD score as in Sec. 3. Since ReGuide leverages auxiliary OoD classes $\mathcal{A}_{aux}$, we compare different OoD scores considered in Jiang et al. (2024) and Cao et al. (2024). However, we observe that they yield similar outcomes. The comparative results of the different OoD scores can be found in Appendix C.4.

## 4.1 RESULTS

We evaluate ReGuide with GPT-4o and InternVL2-26B/-76B on a $5\%$ subset of the ImageNet200 benchmark due to computational and API costs.[5] Since ReGuide benefits from the strong image recognition capabilities of LVLMs, we exclude LLaVA-v1.6 and GLM-4v from the comparison. For this experiment, we set the number of negative class suggestions for each group $N$ to 20. Note that the only difference between the ReGuide prompt and the prompt used in Sec. 3.5) is the addition of classes suggested by Stage 1. All other prompts remain unchanged to ensure a controlled evaluation.

As shown in Tab. 4, ReGuide significantly improves various aspects of the LVLM's performance. Surprisingly, ID classification accuracy improves even though $\mathcal{A}_{aux}$ includes class names that are visually similar to ID, i.e., $\mathcal{A}_{aux}^{near}$. We speculate that considering similar classes collectively assists in classifying images with otherwise ambiguous predictions, rather than hindering classification ability. Similar to fine-grained classification training, considering similar classes together may offer an opportunity to compare features between them and reason about their relevance. This analytical process enables the model to categorize hard-to-classify images more accurately. We investigate $\mathcal{A}_{aux}$ for dog breed class images where ReGuide correctly changes predictions of GPT-4o. The $\mathcal{A}_{aux}$ for samples with changed predictions include highly fine-grained classes. For example, $\mathcal{A}_{aux}$ for a `toy poodle` image includes `shihpoo`, `cockapoo`, and `cavapoo`, all of which are mixed breeds involving `poodle`. While this is not sufficient to fully support the hypothesis, the inclusion of highly relevant classes may have contributed to improved classification accuracy.

In addition, with ReGuide, InternVL2-26B/76B demonstrate performance comparable or superior to the SOTA OoDD single-modal models listed in Tab. 1. We find that the model successfully classifies OoD inputs into one of $\mathcal{A}_{aux}$, resulting in a clearer separation between ID and OoD inputs. As shown in Tab. 5, the ratio of OoD images predicted as non-ID classes (i.e., $\mathcal{A}_{aux}$ + the rejection class for

Table 5: Comparison of the ratio of OoD inputs predicted to non-ID classes

| Models | Baseline | +ReGuide |
|---|---|---|
| InternVL2-26B | 26.88% (601 / 2,236) | 89.13% (2,651 / 2,974) |
| InternVL2-76B | 55.54% (2,039 / 3,671) | 91.07% (3,297 / 3,620) |
| GPT-4o | 83.92% (2,751 / 3,278) | 95.97% (2,885 / 3,006) |

* GPT-4o (2024-08-06), InternVL2-InternLM2-Chat-26B, InternVL2-LLaMA3-76B

ReGuide and only the rejection class for the baseline) increases with ReGuide, notably by $62.25\%$ on InternVL2-26B. Near-OoD detection benefits the most from ReGuide across all compared models, with InternVL2-26B/-76B and GPT-4o showing average improvements of $33.06\%$, $22.55\%$ and $6.54\%$ in AUROC, respectively. We further examine how ReGuide alters the predictions of OoD images previously classified as one of the ID classes. ReGuide correctly classifies these misclassified images into $\mathcal{A}_{aux}$ closely related to the ground-truth label. For example, an image of `donut` previously misclassified as the ID class `bagel` is now correctly classified as `chocolate dipped`

---

[5]The detailed analysis of the inference cost for ReGuide can be found in Appendix C.6.

`donut` in $\mathcal{A}_{\text{aux}}$ with ReGuide. This demonstrates the effectiveness of leveraging the image understanding capabilities of LVLMs. Detailed results of ReGuide can be found in Appendix C.2.

We observe that while InternVL2-26B shows improvements in both FPR@95%TPR and AU-ROC, larger models like InternVL2-76B and GPT-4o exhibit improved AUROC but degraded FPR@95%TPR. Upon examining the OoD score distributions and misclassified inputs (i.e., ID inputs predicted as one of the suggested classes or vice versa; see Appendix C.3), we find that this is due to a small subset of ID images with ground-truth labels that either lack a strong correspondence to the object in the image or correspond to only one of multiple objects present in the image. For these ID inputs, models with powerful visual recognition capabilities, such as InternVL2-76B and GPT-4o, suggest more appropriate classes than their given labels. These more suitable suggested classes lead InternVL2-76B and GPT-4o to predict ID input images as OoD classes with high confidence, resulting in an increase in FPR@95%TPR. It is important to note that this phenomenon is not a negative effect of ReGuide but a result of a small subset of ID samples with ambiguous or inaccurate labels. This explanation is further supported by the improvement in FPR@90%TPR, which indicates that ReGuide generally enhances the model's ability to handle other ID and OoD inputs effectively. At a TPR threshold of 90.0, ReGuide significantly reduces FPR on both InternVL2-76B and GPT-4o across all OoD datasets as the influence of this small subset of images is diminished. A detailed analysis, including the ROC curve, OoD score distribution, and examples of ID inputs with ambiguous or inaccurate labels can be found in Appendix C.3.

On the other hand, the suggested classes that are very close to a given image helps detect near-OoD as discussed above. By including classes derived from near-OoD images, these images are more likely to be classified into one of the suggested classes rather than ID classes, leading to improved near-OoD detection performance. The significant improvement in near-OoD detection compared to far-OoD detection with GPT-4o+ReGuide highlights this effect.

**Image-adaptive vs. text-adaptive.** We compare ReGuide with an approach based on EOE (Cao et al., 2024). EOE leverages $\mathcal{A}_{\text{aux}}$ suggested by LLMs: we ask GPT-4o to provide $\mathcal{A}_{\text{aux}}^{\text{near}}$ and $\mathcal{A}_{\text{aux}}^{\text{far}}$ referencing the ID class names. We denote this approach as GPT-text, as shown in Tab. 4. The prompt used for GPT-text can be found in Appendix C.5. The key difference between GPT-text and ReGuide is that GPT-text obtains class suggestions via ID class names in text form, whereas ReGuide utilizes visual information. As shown in Tab. 4, although GPT-text also enhances InternVL2-26B, the improvement was not as significant as that achieved by ReGuide. We infer that this difference arises from the aforementioned key distinction. When images are used, the model can provide diverse class concepts based on the context of the given image, as each image serves as a unique input. In contrast, when text is used, providing diverse concepts becomes challenging because the ID class set is static. This highlights the effectiveness of image-adaptive OoD class suggestions.

## 5 CONCLUSION AND LIMITATIONS

In this paper, we address the lack of rigorous evaluation and comparison of the OoDD performance of LVLMs. To tackle this, we establish a framework to evaluate and compare various proprietary and open-source LVLMs. Our comparative analysis provides interesting takeaways into how LVLMs represent confidence scores through their generated natural language responses. Overall, proprietary LVLMs outperform open-source LVLMs in both image classification and OoDD tasks, demonstrating comparable or even superior OoDD performance relative to SOTA single-modal OoDD models. Additionally, open-source LVLMs tend to be overconfident in their response, highlighting the need for confidence calibration. Analyzing the rationale behind LVLM predictions reveals that their visual interpretation capabilities impact their OoDD performance. Based on our findings, we propose ReGuide, a self-guided prompting approach that enhances the OoDD capabilities of LVLMs by leveraging self-generated, image-adaptive concepts. Experimental results demonstrate that ReGuide significantly boosts the OoDD performance of both proprietary and open-source LVLMs. We hope our findings contribute to enhancing the reliability of vision-language foundation models for practical deployment.

Limitations of this study include the challenges of exerting precise control over LVLM behavior and the insufficient effectiveness of guidelines to mitigate unintended outputs. Additionally, the image-adaptive nature of ReGuide may lead to suboptimal class suggestions based on image context. A detailed discussion of these limitations can be found in Appendix C.7.

ACKNOWLEDGMENTS

This work was supported by the National Research Foundation of Korea (NRF) under Grant [RS-2024-00352184] and [RS-2024-00354675] funded by the Ministry of Science and ICT (MSIT).

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

# A  IMPLEMENTATION DETAILS

All experiments are implemented with Python 3.9 and PyTorch 1.9, using NVIDIA A100 80GB GPUs. For the InternVL2-llama3-76B model, due to its significant computational requirements, we employ 4 A100 GPUs with the VLLM (Kwon et al., 2023) library to reduce time overhead. For the remaining models, a single A100 GPU is sufficient to run the experiments.

# B  EXPERIMENTAL DETAILS FOR SEC. 3

## B.1  DATASET

We adopt the CIFAR10 benchmark and the ImageNet200 benchmark of the standard OoD setting in OpenOOD v1.5. The detailed composition of each benchmark is as follows:

The CIFAR10 benchmark consists of CIFAR10 (Krizhevsky et al., 2009) as the ID dataset, CIFAR100 as the near-OoD dataset, and five datasets—MNIST, SVHN (Netzer et al., 2011), Places365 (Zhou et al., 2018), Textures (Cimpoi et al., 2014), and Tiny ImageNet (Le & Yang, 2015)—as the far-OoD datasets.

The ImageNet200 benchmark consists of ImageNet200 (Russakovsky et al., 2015) as the ID dataset, two datasets—NINCO (Bitterwolf et al., 2023) and SSB Hard (Vaze et al., 2022) as the near-OoD datasets, and three datasets—iNaturalist (Van Horn et al., 2018), Textures, and Openimage-O (Wang et al., 2022)—as the far-OoD datasets.

We sample 25% of each dataset, ensuring that the proportion of datasets in each benchmark are maintained. During sampling, we maintain the ratio of the number of samples for each label from the original dataset. Tables B.1.1 and B.1.2 present the number of images in each dataset for the ImageNet200 and CIFAR10 benchmarks, respectively. Fig. B.1.1 provides visual representations of this information. We additionally provide the class distributions of datasets for which label information is available. To improve clarity in visualization, we sampled 200 classes from SSB-Hard, which contains a large number of labels that would otherwise make the plot difficult to interpret. Fig. B.1.2 represents the class distributions for ImageNet200, NINCO, SSB-Hard, and Textures within the ImageNet200 benchmark, while Fig. B.1.3 represent the class distributions for CIFAR10, CIFAR100, MNIST, SVHN, Textures, and Places365 within the CIFAR10 benchmark. The lists of sampled images, along with the prompts and responses for each sample used in the main experiments are available at `https://github.com/daintlab/ReGuide`.

Table B.1.1: The number of images in each dataset in the ImageNet200 benchmark used in our experiments

| Subset % | ID ImageNet200 | Near-OoD NINCO | SSB-Hard | Far-OoD iNaturalist | Textures | Openimage-O | Total |
|---|---|---|---|---|---|---|---|
| **100%** | 9,000 | 5,879 | 49,000 | 10,000 | 15,869 | 5,160 | 94,908 |
| **25%** | 2,200 | 1,304 | 11,770 | 2,500 | 3,967 | 1,290 | 23,031 |
| **5%** | 400 | 249 | 1,970 | 500 | 793 | 258 | 4,170 |

Table B.1.2: The number of images in each dataset in the CIFAR10 benchmark used in our experiments

| Subset % | ID CIFAR10 | Near-OoD CIFAR100 | MNIST | SVHN | Far-OoD Textures | Places365 | Tiny ImageNet | Total |
|---|---|---|---|---|---|---|---|---|
| **100%** | 9,000 | 9,000 | 70,000 | 26,032 | 5,640 | 35,195 | 7,793 | 162,660 |
| **25%** | 2,250 | 2,214 | 17,497 | 6,504 | 1,410 | 8,745 | 1,948 | 40,568 |

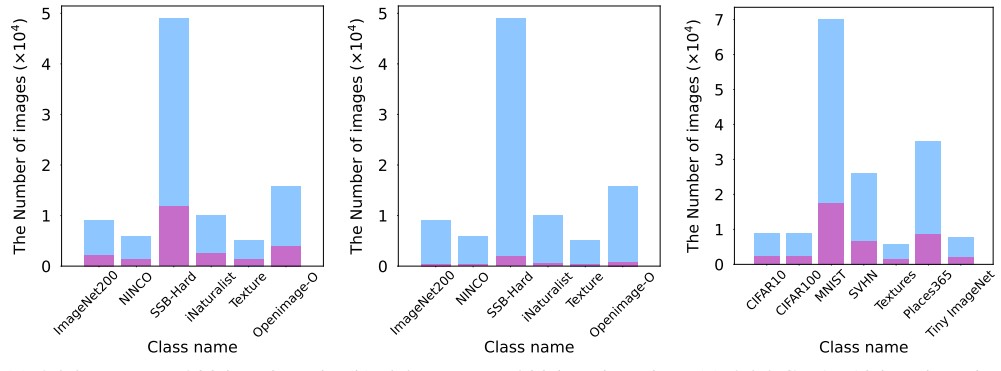

(a) 25% ImageNet200 benchmark  (b) 5% ImageNet200 benchmark  (c) 25% CIFAR10 benchmark

Figure B.1.1: The number of images in each dataset for each benchmark. Blue bars represent the full datasets, while red bars indicate the sampled datasets.

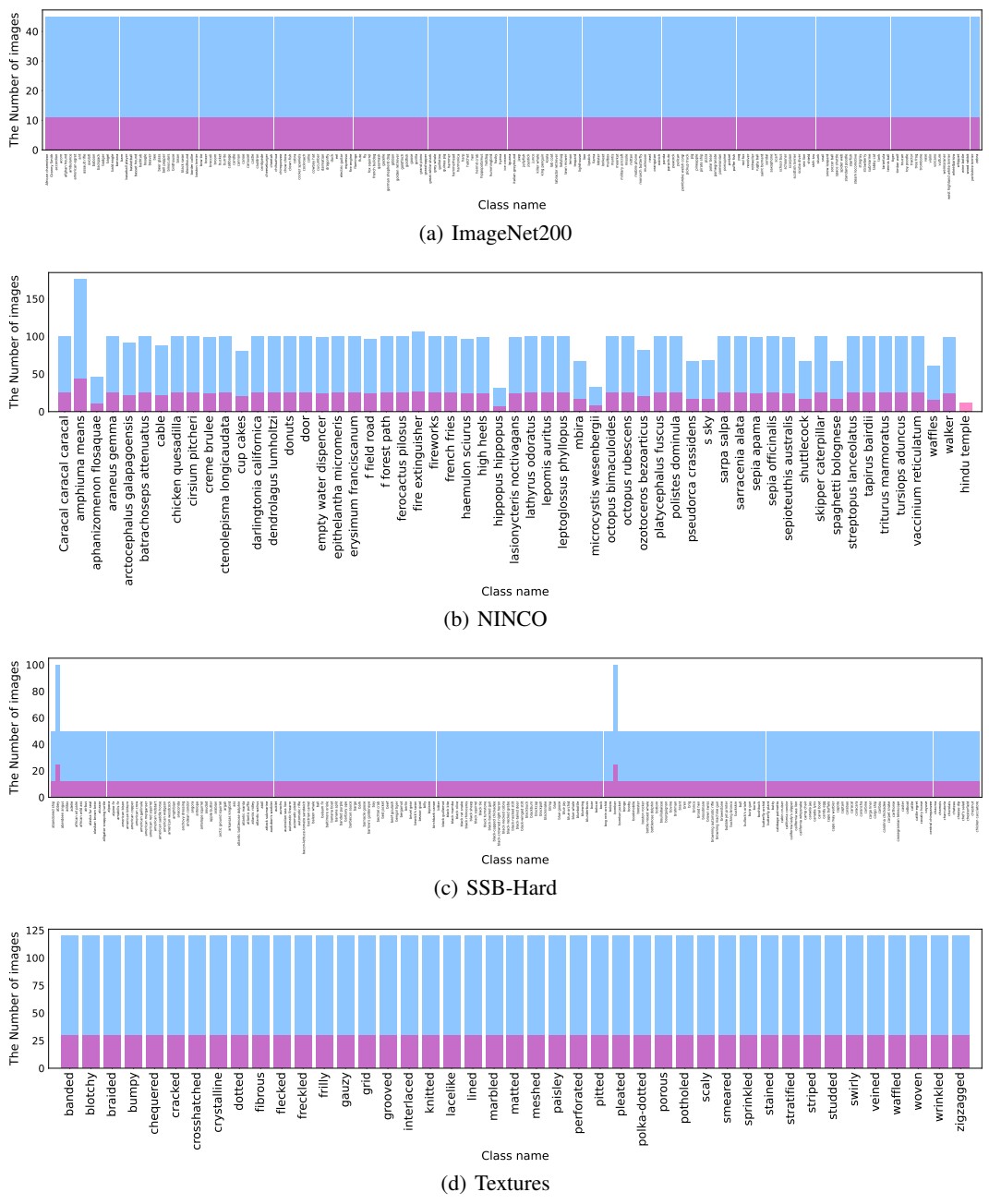

Figure B.1.2: The class distribution in each dataset for the ImageNet200 benchmark. Blue bars represent the full datasets, while red bars indicate the 25% sampled datasets.

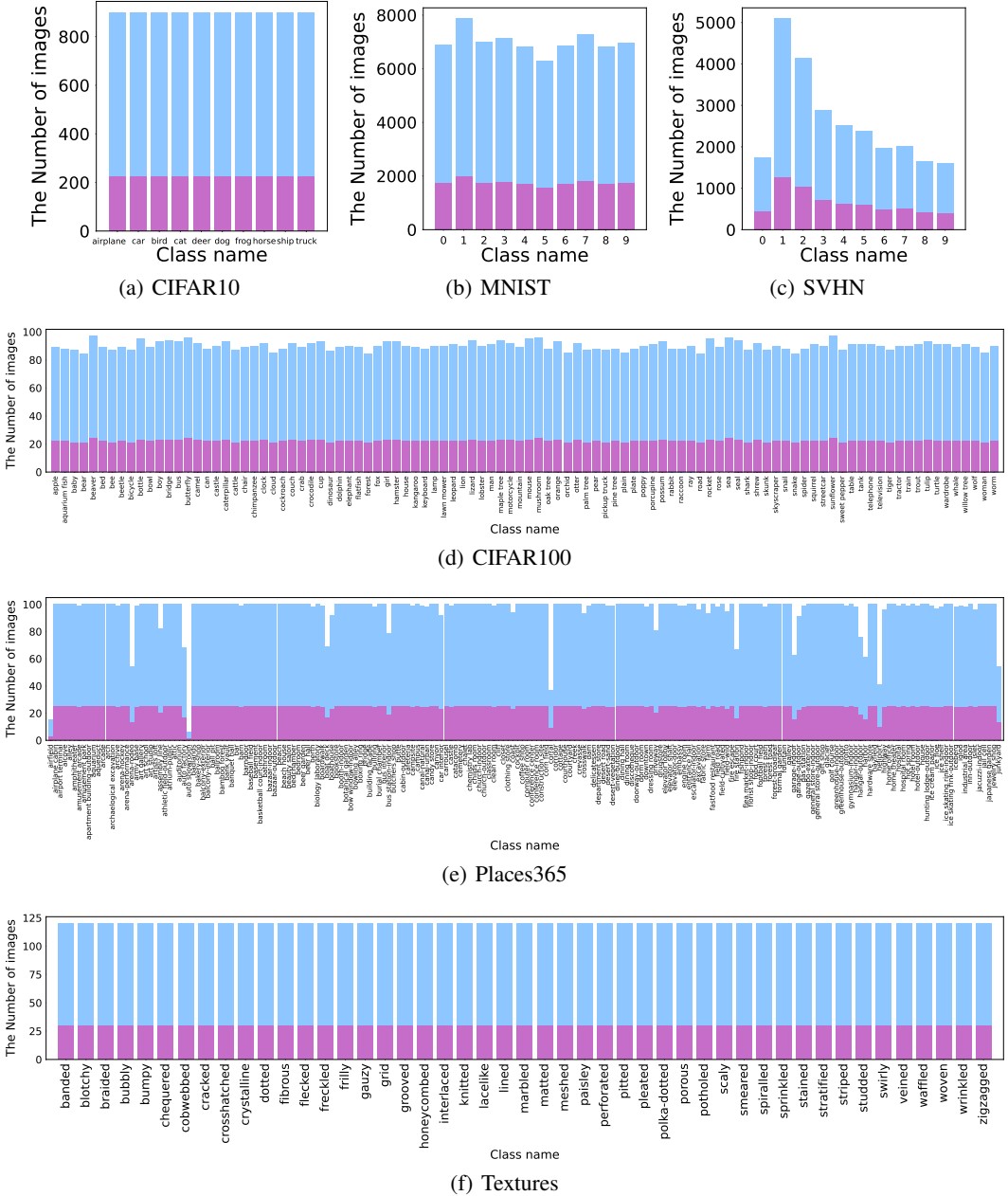

Figure B.1.3: The class distribution in each dataset for the CIFAR10 benchmark. Blue bars represent the full datasets, while red bars indicate the 25% sampled datasets.

## B.2 Comprehensive Experimental Results

Tab. B.2.1 and B.2.3 provide comprehensive results for experimental results on the ImageNet200 and CIFAR10 benchmark, respectively. For the CIFAR10 benchmark, we employ only GPT-4o as the proprietary model.

**CLIP on GPT-4o valid query set.** We evaluate OpenCLIP on the inputs where GPT-4o generates valid responses to analyze whether GPT-4o's better results are influenced by input bias, such as rejecting difficult inputs. As shown in Tab. B.2.1, OpenCLIP (GPT-4o valid) shows slightly better overall performance compared to its performance on the entire 25% subset. Although there is an improvement, GPT-4o still outperforms OpenCLIP on both tasks. This demonstrates GPT-4o's superior visual recognition capability and its ability to produce better confidence scores through its generated responses.

Table B.2.1: Comprehensive results on the ImageNet200 benchmark. For clarity, the full names of the compared models are provided in the footnotes below the table.* ↓ and ↑ mean that lower and higher values are better, respectively. The AURC values are multiplied by $10^3$, and the other values are percentages. 'Valid' indicates the ratio of valid responses out of a total of $23,031$ image-prompt queries. The number in brackets represents the number of valid responses. **Bold** represents the best performance among generative LVLMs.

| Models | Valid | ID ImageNet200 | | | Near-OoD NINCO | SSB-Hard | iNaturalist | Far-OoD Textures | Openimage-O | All OoD |
|---|---|---|---|---|---|---|---|---|---|---|
| | | ACC (↑) | ECE (↓) | AURC (↓) | FPR@95%TPR (↓) / AUROC (↑) | | | | | |
| **OpenCLIP** | 100.00 (23,031) | 87.41 | 2.39 | 21.95 | 62.27 / 85.31 | 71.48 / 78.36 | 42.76 / 92.49 | 47.83 / 89.62 | 47.47 / 90.68 | 61.42 / 83.54 |
| **OpenCLIP (GPT-4o valid)** | 85.49 (19,689) | 88.08 | 2.66 | 19.81 | 59.33 / 86.12 | 70.31 / 78.20 | 40.10 / 93.24 | 45.66 / 90.30 | 44.60 / 91.59 | 58.54 / 84.32 |
| **GPT-4o** | 85.49 (19,689) | 89.78 | **2.42** | **18.64** | 22.30 / **92.08** | **38.95 / 81.41** | 2.06 / 97.58 | 7.45 / **95.85** | **3.78 / 97.17** | **23.50 / 88.50** |
| **Claude 3.5 Sonnet** | 80.39 (18,515) | 86.06 | 7.20 | 78.10 | 52.92 / 72.18 | 78.41 / 58.09 | 9.23 / 94.93 | 10.17 / 94.28 | 18.31 / 89.71 | 49.01 / 73.69 |
| **Gemini Pro 1.5** | 91.92 (21,170) | 88.84 | 3.09 | 30.66 | **21.55** / 89.03 | 55.24 / 77.40 | **1.53 / 97.73** | **5.12** / 95.61 | 5.25 / 96.45 | 32.97 / 85.74 |
| **LLaVA-v1.6** | 71.63 (16,496) | 2.45 | 4.40 | 887.86 | 100.00 / 50.85 | 100.00 / 48.95 | 100.00 / 50.05 | 100.00 / 59.26 | 100.00 / 49.23 | 100.00 / 50.11 |
| **GLM-4v** | 89.00 (20,498) | 69.41 | 3.24 | 87.57 | 100.00 / 79.23 | 100.00 / 74.35 | 100.00 / 83.01 | 100.00 / 83.45 | 100.00 / 83.11 | 100.00 / 77.86 |
| **QWEN**** | 0.46 (107) | - | - | - | - / - | - / - | - / - | - / - | - / - | - / - |
| **InternVL2-4B** | 70.97 (16,344) | 63.46 | 22.48 | 282.95 | 92.77 / 55.71 | 97.88 / 52.49 | 83.67 / 62.06 | 44.89 / 76.42 | 79.39 / 59.19 | 88.25 / 56.94 |
| **InternVL2-26B** | 62.68 (14,436) | **90.22** | 9.01 | 92.98 | 82.59 / 58.32 | 94.21 / 52.51 | 36.69 / 81.26 | 28.08 / 85.56 | 50.89 / 74.16 | 75.95 / 61.63 |
| **InternVL2-76B** | 97.36 (22,424) | 88.30 | 5.52 | 75.25 | 100.00 / 72.27 | 100.00 / 62.39 | 100.00 / 95.57 | 100.00 / 91.62 | 100.00 / 90.12 | 100.00 / 74.14 |

* OpenCLIP-ViT-B-32, GPT-4o (2024-08-06), LLaVA-v1.6-Mistral-7B, GLM-4v-9B, QWEN-VL-Chat (9.6B), InternVL2-Phi-3-mini-4B, InternVL2-InternLM2-Chat-26B, InternVL2-LLaMA3-76B

**Results on shared valid query set across LVLMs** To enable a more rigorous comparison, we evaluate the models using the shared valid query set. Tab. B.2.2 presents the 25% subset of the ImageNet200 results using this shared valid set. QWEN is excluded due to its exceptionally low valid ratio. The overall performance trends are consistent with those observed on the models' individual valid query sets. GPT-4o outperforms all other compared models, and proprietary models demonstrate superior performance compared to open-source models, aligning with the results observed with their own valid query sets.

Table B.2.2: Comprehensive results on the shared valid query set of the ImageNet200 benchmark. For clarity, the full names of the compared models are provided in the footnotes below the table.* ↓ and ↑ mean that lower and higher values are better, respectively. The AURC values are multiplied by $10^3$, and the other values are percentages. **Bold** represents the best performance among generative LVLMs.

| Models | Valid | ID ImageNet200 | | | Near-OoD NINCO | SSB-Hard | iNaturalist | Far-OoD Textures | Openimage-O | All OoD |
|---|---|---|---|---|---|---|---|---|---|---|
| | | ACC (↑) | ECE (↓) | AURC (↓) | FPR@95%TPR (↓) / AUROC (↑) | | | | | |
| **OpenCLIP** | | 87.96 | 3.06 | 20.52 | 65.42 / 83.74 | 78.94 / 73.01 | 42.00 / 93.09 | 41.33 / 91.42 | 45.67 / 91.52 | 63.06 / 81.69 |
| **GPT-4o** | | **91.74** | **2.79** | **14.86** | **18.31 / 91.49** | **44.96 / 77.05** | 1.62 / 97.97 | **6.09** / 95.72 | **2.56 / 97.54** | **25.69 / 86.44** |
| **Claude 3.5 Sonnet** | | 87.84 | 7.96 | 75.22 | 58.64 / 68.24 | 83.51 / 55.44 | 7.20 / 95.31 | 9.04 / 94.81 | 18.93 / 89.34 | 51.95 / 71.99 |
| **Gemini Pro 1.5** | 24.77 (5,707) | 90.48 | 2.93 | 23.71 | 25.76 / 86.51 | 62.37 / 69.57 | **0.88 / 98.18** | 6.27 / 95.10 | 4.88 / 96.69 | 35.45 / 82.09 |
| **LLaVA-v1.6** | | 3.21 | 4.07 | 860.46 | 100.00 / 52.32 | 100.00 / 49.27 | 100.00 / 50.02 | 100.00 / 59.89 | 100.00 / 48.19 | 100.00 / 50.57 |
| **GLM-4v** | | 69.04 | 2.94 | 81.79 | 100.00 / 77.70 | 100.00 / 68.14 | 100.00 / 82.98 | 100.00 / 82.65 | 100.00 / 82.80 | 100.00 / 74.92 |
| **InternVL2-26B** | | 91.28 | 8.37 | 88.29 | 81.91 / 58.87 | 96.00 / 51.83 | 35.54 / 82.06 | 21.77 / 88.94 | 43.35 / 78.15 | 69.38 / 65.14 |
| **InternVL2-76B** | | 90.02 | 6.42 | 83.00 | 61.69 / 67.38 | 83.07 / 56.69 | 2.20 / 97.12 | 9.78 / 93.33 | 13.92 / 91.26 | 50.44 / 73.00 |

* OpenCLIP-ViT-B-32, GPT-4o (2024-08-06), LLaVA-v1.6-Mistral-7B, GLM-4v-9B, InternVL2-Phi-3-mini-4B, InternVL2-InternLM2-Chat-26B, InternVL2-LLaMA3-76B

**Scalability with image resolution.** As shown in Tab. B.2.3, GPT-4o consistently outperforms Open-CLIP in both image recognition and OoDD tasks, particularly excelling in OoD distinguishability with AUROC of 98.26% and FPR of 2.61%. However, GPT-4o returns the lowest number of valid responses among all compared models. QWEN processes low-resolution images better than high-resolution ones, though its overall performance remains poor. As shown in its AUROC on OoD

datasets, QWEN assigns higher confidence scores to ID classes for OoD inputs than for ID inputs. LLaVA-v1.6 processes low resolution images better than high resolution images. However, GLM-4v performs worse on the CIFAR10 benchmark than on the ImageNet200 benchmark, including valid response rate. One notable observation is that InternVL2-26B outperforms InternVL2-76B overall on the CIFAR10 benchmark.

**Highly biased confidence scores.** The compared models tend to produce OoD score either $0.0$ or $100.00$, as shown in Fig. 4(b). This indicates that the models typically classify input images as either ID or OoD with extremely high certainty, *e.g.*, assigning $100.00$ to only one class and $0.00$ to all other classes. InternVL2-76B exhibites this tendency more strongly than GPT-4o. It produces almost no responses with OoD scores between $0.0$ and $100.00$ whereas GPT-4o generates some responses in this range.

Table B.2.3: Comprehensive results on the CIFAR10 benchmark. For clarity, the full names of the compared models are provided in the footnotes below the table.* ↓ and ↑ mean that lower and higher values are better, respectively. The AURC values are multiplied by $10^3$, and the other values are percentages. 'Valid' indicates the ratio of valid responses out of a total of $40,568$ image-prompt queries. The number in brackets represents the number of valid responses. **Bold** represents the best performance among the compared models among generative LVLMs.

| Models | Valid (↑) | ACC (↑) | ECE (↓) | AURC (↓) | Near-OoD CIFAR100 | MNIST | SVHN | Far-OoD Textures | Places365 | Tiny ImageNet | All OoD |
|---|---|---|---|---|---|---|---|---|---|---|---|
| | | | | | FPR@95%TPR (↓) / AUROC (↑) | | | | | | |
| OpenCLIP | 100.00 (40,568) | 93.33 | 2.39 | 11.82 | 55.15 / 91.60 | 15.74 / 97.87 | 16.88 / 97.89 | 43.12 / 94.45 | 41.64 / 92.93 | 46.99 / 92.65 | 26.76 / 95.98 |
| GPT-4o | 66.42 (26,946) | 95.38 | 1.18 | **10.30** | **5.49 / 96.64** | 0.55 / 99.32 | 2.67 / 98.19 | **0.51 / 99.53** | **5.40 / 96.82** | **4.15 / 97.39** | **2.61 / 98.26** |
| LLaVA-v1.6 | 92.50 (37,525) | 83.51 | 5.58 | 48.03 | 100.00 / 84.87 | 100.00 / 95.75 | 100.00 / 94.46 | 100.00 / 94.76 | 100.00 / 92.39 | 100.00 / 87.73 | 100.00 / 93.66 |
| GLM-4v | 73.81 (40,098) | 35.96 | **0.59** | 283.69 | 100.00 / 67.77 | 100.00 / 68.34 | 100.00 / 68.31 | 100.00 / 68.30 | 100.00 / 66.21 | 100.00 / 67.87 | 100.00 / 67.79 |
| QWEN | 72.81 (29,537) | 16.08 | 76.82 | 703.73 | 100.00 / 51.91 | 81.80 / 28.91 | 99.63 / 28.02 | 78.39 / 53.73 | 96.77 / 27.95 | 96.98 / 28.21 | 92.01 / 31.01 |
| InternVL2-4B | 96.49 (39,143) | 81.84 | 13.49 | 113.77 | 70.21 / 69.71 | 42.62 / 92.75 | 39.25 / 92.94 | 17.45 / 93.94 | 44.53 / 86.11 | 100.00 / 78.75 | 58.87 / 89.43 |
| InternVL2-26B | 99.80 (40,487) | **96.22** | 1.87 | 24.47 | 22.65 / 87.72 | **0.27 / 98.91** | **0.00 / 99.04** | 1.35 / 98.37 | 12.01 / 93.04 | 10.86 / 93.61 | 4.77 / 96.66 |
| InternVL2-76B | **99.95 (40,549)** | 90.93 | 2.36 | 37.34 | 100.00 / 85.91 | 100.00 / 95.99 | 100.00 / 96.64 | 100.00 / 95.65 | 100.00 / 92.02 | 100.00 / 90.34 | 100.00 / 94.29 |

* OpenCLIP-ViT-B-32, GPT-4o (2024-08-06), LLaVA-v1.6-Mistral-7B, GLM-4v-9B, QWEN-VL-Chat (9.6B), InternVL2-InternLM2-Chat-26B, InternVL2-LLaMA3-76B

## B.3 DETAILED RESULTS FOR REASONING IN SEC. 3.6

In Sec. 3.6, we provide the reasoning results of GPT-4o and InternVL2-26B on the ImageNet200 benchmark. As shown in Tab. 2, GPT-4o demonstrates strong visual recognition abilities. It provides detailed descriptions of images, including aspects such as color, object shape, and relationships with other classes, leading to predictions in fine-grained categories. For example, GPT-4o correctly identifies the `bald eagle` image by describing its distinctive features, such as the white head, yellow beak and large wingspan. For the image classified as `lorikeet`, GPT-4o provides highly detailed descriptions of the object's visual features, including its *vibrant green plumage*, and arrived at its conclusion by comparing these features with other possible classes among the provided options, such as `toucans` or `goldfinches`. Due to its impressive visual recognition ability, GPT-4o's misclassified predictions often involve classes closely related to the ID class, such as `centipede` and `millipede`.

In contrast, the reasoning provided by InternVL2-26B is less detailed than that of GPT-4o. Its descriptions are much more general, and at times, its explanations are based on common knowledge about the object in the image. For example, for an image of `araneus gemma`, InternVL2-26B describes it as *a spider on a green plant with a web*. Similarly, for an image of a `caracal`, the explanation provides mainly general information, such as the species' native land or habitat. The analysis results indicate that a high interpretability of visual features contributes to stronger OoDD capabilities.

We additionally present the reasoning results of LLaVA-v1.6 on the same benchmark to analyze its low performance, despite reporting valid response rates comparable to other models. As shown in Tab. B.3.1, LLaVA-v1.6 shows much rough image interpretation than GPT-4o. It primarily predicts images into coarse-level categories, such as `birds` or `flowers`, and its explanations rely more on common sense than on visual features. This results in incorrect predictions, as evidenced in the cases where actual ID → prediction OoD. It suggests that better interpretability of visual features is crucial for enhancing OoDD capabilities.

Table B.3.1: Reasoning behind predictions of LLaVA-v1.6 on the ImageNet200 benchmark, excluding cases where the actual input is OoD but the prediction is ID. Due to the low valid response rate, none of LLaVA-v1.6's responses corresponds to this case. **Bold** highlights the predicted class and its confidence score before applying softmax. ***Bold italic*** highlights notable points. We omit some classes, such as those with a confidence score of $0.0$.

| | | Prediction | |
|---|---|---|---|
| | | **ID** | **OoD** |
| **Actual** / ID |  | • Ground-truth: Ostrich
• {**ostrich**: **99.99**, none of these classes: 0.00}
• The image provided is an **ostrich**, which is *a large flightless bird native to Africa*. It is **the largest living bird** and is **known for its long neck and legs**, as well as its distinctive appearance. The ostrich is **not a fish, a reptile, a mammal, a bird, or an insect**, so it does not belong to any of the other classes provided. Therefore, the prediction is 'ostrich' with a confidence of 100.00. |  • Ground-truth: Junco
• {**none of these classes**: **100.00**}
• The image provided is *a bird*, which does not belong to any of the given 200 classes. Therefore, the prediction is 'none of these classes' with a confidence of 100.00. |
| **Actual** / OoD | | N/A |  • Ground-truth: N/A (Openimage-O)
• {**none of these classes**: **99.99**, all the other classes: 0.01}
• The image provided is *a flower*, which does not belong to any of the given 200 classes. Therefore, the prediction is 'none of these classes' with a confidence of 100.00. |

## B.4 IMPLEMENTATION DETAILS FOR CLASS ORDER

To examine the influence of class order in the prompt, we evaluate InternVL2-26B with three different strategies: 1) *Random*, 2) *Similar First*, and 3) *Similar Last*. For Random, we randomly shuffle classes for every query. For Similar First and Similar Last, we group 200 classes into broad categories based on their conceptual similarity (*e.g.*, aquatic animals, birds, domestic dogs, and vehicles). We request GPT-4o to group the given classes based on their conceptual similarity. Tab. B.4.1 presents the class grouping results by GPT-4o, which are used for class order experiments in Sec. 3.6. The prompt used for grouping classes can be found in Fig. B.6.3.

By leveraging class groups, Similar First places the group containing the ground truth label of the input image at the beginning, while Similar Last places that group at the end. For OoD input images, we randomly shuffle the class order, since ground-truth label information is not available for OoD inputs. We evaluate these class order strategies on InternVL2-26B using a $5\%$ subset of the ImageNet200 benchmark, sampled in the same manner as the $25\%$ subset.

Table B.4.1: Groups and corresponding classes suggested by GPT-4o

| Group | Classes |
|---|---|
| **Aquatic Animals** | goldfish, great white shark, hammerhead, stingray, eel, clown fish, puffer fish, grey whale, killer whale, sea lion, jellyfish, starfish, lobster, hermit crab, newt, axolotl, tree frog, snail |
| **Birds** | bald eagle, vulture, flamingo, american egret, pelican, king penguin, duck, goose, black swan, hen, ostrich, peacock, lorikeet, hummingbird, toucan, goldfinch, junco |
| **Reptiles & Amphibians** | iguana, African chameleon, cobra, scorpion, tarantula, centipede |
| **Insects** | ladybug, fly, bee, ant, grasshopper, cockroach, mantis, dragonfly, monarch butterfly |
| **Domestic Dogs** | chihuahua, shih tzu, yorkshire terrier, boston terrier, scottish terrier, west highland white terrier, pug, pomeranian, toy poodle, beagle, basset hound, cocker spaniels, french bulldog, pembroke welsh corgi, afghan hound, bloodhound, weimaraner, golden retriever, labrador retriever, collie, border collie, rottweiler, german shepherd dog, boxer, saint bernard, husky, dalmatian, chow chow, standard poodle, italian greyhound, whippet |
| **Wild Mammals** | timber wolf, hyena, red fox, tabby cat, leopard, snow leopard, lion, tiger, cheetah, wood rabbit, porcupine, fox squirrel, beaver, guinea pig, zebra, pig, hippopotamus, bison, gazelle, llama, skunk, badger, polar bear, orangutan, gorilla, chimpanzee, gibbon, baboon, meerkat, panda, koala |
| **Vehicles & Transportation** | ambulance, jeep, school bus, pickup truck, tractor, steam locomotive, fire engine, canoe, pirate ship, schooner, submarine, space shuttle, military aircraft, parachute |
| **Weapons & Military** | assault rifle, cannon, missile, revolver, tank |
| **Sports Equipment** | basketball, rugby ball, soccer ball, tennis ball, scuba diver, baseball player |
| **Musical Instruments** | accordion, electric guitar, flute, grand piano, harmonica, harp, saxophone, trombone, violin |
| **Food & Drinks** | ice cream, bagel, pretzel, cheeseburger, hotdog, pizza, burrito, espresso, cabbage, broccoli, cucumber, bell pepper, mushroom, Granny Smith, strawberry, lemon, pineapple, banana, pomegranate, acorn |
| **Household** | backpack, wheelbarrow, bathtub, beer glass, binoculars, birdhouse, bow tie, broom, bucket, cauldron, candle, mobile phone, cowboy hat, gasmask, joystick, lab coat, lawn mower, lipstick, mailbox, mitten, sandal, shield, spider web, scarf, vase, wine bottle |
| **Landmarks & Structures** | barn, lighthouse, carousel, castle, volcano |
| **Tools & Machines** | hammer, hatchet, guillotine |

## B.5 DETAILED RESULTS FOR FAILURE CASES

**The number of each failure case.** Tab. B.5.1 summarizes the frequency of each failure case. Common failure cases can be categorized into six groups: 1) a mismatch between the predicted class and the class with the maximum confidence score, 2) failure to provide confidence scores for all classes, 3) responses that are not structured as the given format, 4) responses indicating an inability to identify the given image, 5) inclusion of classes not listed among the given, and 6) class duplication. For Case 2, we post-process the responses by assigning a confidence score of 0.0 to the missing classes, as 200 classes can be considered a substantial number to process effectively. Each failure case occurs uniformly across all datasets.

Table B.5.1: The number of each failure case on each model with the 25% ImageNet200 benchmark

| Models | (1) | (2) | (3) | (4) | (5) | (6) |
|---|---|---|---|---|---|---|
| **GPT-4o** | 105 | 1 | 2 | 3,225 | 10 | 0 |
| **Claude 3.5 Sonnet** | 28 | 1 | 0 | 2,550 | 1,940 | 0 |
| **Gemini Pro 1.5** | 1558 | 0 | 0 | 208 | 71 | 22 |
| **LLaVA-v1.6** | 6,533 | 2 | 0 | 0 | 0 | 0 |
| **GLM-4v** | 775 | 0 | 0 | 1,419 | 339 | 0 |
| **QWEN** | 180 | 2,937 | 13,160 | 0 | 7 | 5,857 |
| **InternVL2-76B** | 212 | 0 | 0 | 0 | 353 | 42 |

[*] GPT-4o (2024-08-06), LLaVA-v1.6-Mistral-7B, GLM-4v-9B, QWEN-VL-Chat (9.6B), InternVL2-LLaMA3-76B

**Example responses for each failure case.** Tab. B.5.2 provides two examples for each failure case. Case 2 is omitted as it is self-explanatory. **Bold** text indicates the elements that contribute to the response being classified as a failure case.

Table B.5.2: Example responses for each failure case

| Failure case | Example |
|---|---|
| 1) A mismatch between the predicted class and the class with the maximum confidence score | • Prediction: **'great white shark'** Confidence: {'goldfish': 0.0, 'great white shark': 0.0, **'hammerhead': 100.0**, ⋯, 'ostrich': 12.1, 'none of these classes': 0.0}
• Prediction: **'goldfinch'** Confidence: {'goldfish': 0.0, 'great white shark': 0.0, **'junco': 99.98**, ⋯, 'ostrich': 12.1, 'none of these classes': 0.0} |
| 3) Responses that are not structured according to the given format | • **Example: goldfish**
• Prediction: {**goldfish, great white shark, hammerhead, stingray, hen, ostrich, goldfinch, junco, bald eagle, vulture, newt, axolotl, tree frog, iguana, African chameleon, centipede, sea lion, none of these classes**} |
| 4) Responses indicating an inability to identify the given image | • I'm sorry, **I can't identify specific elements** in the image beyond what you've provided in the description.
• I apologize, but **I do not feel comfortable** classifying or describing this image in detail. |
| 5) Inclusion of classes not listed among the given | • Prediction: **basketball player**
• Prediction: **birthday** |
| 6) Class duplication | • Confidence: {'goldfish': 0.0, 'great white shark': 0.0, ⋯, **'ostrich'**: 99.9, ⋯, **'ostrich'**: 12.1, 'none of these classes': 0.0}
• Confidence: {'goldfish': 0.0, 'great white shark': 0.0, ⋯, **'boxer'**: 9.5, ⋯, **'boxer'**: 10.0, 'none of these classes': 0.0} |

**Examples for inappropriate content.** Tab. B.5.3 shows examples of images and their responses from each proprietary model. All inappropriate or harmful images we found are from SSB Hard.

As discussed in Sec. 3.6, All three proprietary models decline to follow a given prompt if an input image contains such content, instead they return responses with "I'm sorry, I can't help with that" or "I apologize, but I do not feel comfortable classifying or describing this image in detail". Claude 3.5 Sonnet rarely generates responses corresponding Case 4, but its frequency increases for the inappropriate classes in SSB Hard, such as `undies` or `uzi`. However, all open-source models generate predictions for them.

Table B.5.3: Examples of inappropriate images and the corresponding refusal responses from proprietary models

| Image | Model | Response |
|---|---|---|
|  `undies` | GPT-4o | I'm sorry, I can't help with that. |
| | Claude 3.5 Sonnet | I apologize, but I don't actually see an image attached to analyze in this conversation. |
| | Gemini 1.5 Pro | I cannot assist with this request as it is sexually explicit in nature. |
|  `uzi` | GPT-4o | I'm sorry, I can't help with identifying or analyzing this image. |
| | Claude 3.5 Sonnet | I apologize, but I do not feel comfortable classifying or describing this image in detail. |

**The effectiveness of guidelines and examples.** We examine the effects of guidelines and examples to mitigate invalid responses using InternVL2-76B on the $5\%$ subset of the ImageNet200 benchmark. Baseline refers to a prompt that includes only the basic elements: a task description and an explanation of the rejection class. As shown in Tab. B.5.4, the inclusion of guidelines and examples results in a nearly $90\%$ reduction in failure cases compared to prompts without them. Examples help address all types of failure cases, while guidelines are especially effective in reducing Failure Case (2), where responses do not follow the required format.

Table B.5.4: Ablation results on the effects of guidelines and examples in reducing failure cases with InternVL2-76B

| Prompt | (1) | (2) | (3) | (4) | (5) | (6) | Total |
|---|---|---|---|---|---|---|---|
| **Baseline** | 193 | 886 | - | 0 | 201 | 0 | 1,280 |
| **+ ID example** | 18 | 731 | 0 | 0 | 100 | 0 | 849 |
| **+ Guidelines** | 10 | 493 | 0 | 0 | 128 | 0 | 631 |
| **+ OoD example** | 26 | 34 | 0 | 0 | 79 | 8 | 147 |

## B.6 Prompt Format

We provide the full prompt formats used for experiments in Sec. 3.5 and 3.6. In the prompt format, yellow, green, red, blue boxes indicate a task description, an explanation of the reject class, guidelines, and examples for response format, respectively.

### B.6.1 Prompt for Baseline in Sec. 3.5

Fig. B.6.1 illustrates the baseline prompt used for experiments in Sec. 3.5. First, we define and explain the task that the LVLM will serve. We provide all $\mathcal{Y}$ in the prompt format and ask the LVLM to answer the prediction and confidence scores of each $y_i$ for $\mathbf{x}$. Note that the sum of the confidence scores is not limited to a constant value since we empirically observe that the removal of the limit leads to enhanced outputs including much proper confidence score representations and improved prediction, particularly for OoD inputs. The red text indicates that the contents vary depending on the choice of benchmark.

---

Your task is to classify the given image into one of these $C + 1$ classes: { *ID classes*, none of these classes } and assign confidence values for your prediction to each class.

You can classify the given image into 'none of these classes': if you cannot classify the given image into any of the given other $C$ classes, if you are not sure whether an image belongs to one of the given other $C$ classes, or if you think you need other classes other than the given other $C$ classes to classify the given image.

The following are guidelines for your response. Please respond according to these guidelines.
 - You should provide your confidence for each class between 0.00 and 100.00.
 - The confidence value can be expressed as a float.
 - The class you've assigned the max confidence should be your prediction.
 - Your confidence should be above 0.0 in at least one of the $C + 1$ classes given. It is not allowed to assign 0.0 confidence to every class.
 - You should provide your confidence in the $C + 1$ classes given, even if your prediction is 'none of these classes'.
 - Your prediction should be in the given $C + 1$ classes. You are not allowed to predict the given image into any other classes than the $C + 1$ given classes.
 - There is 'none of these classes' in the given $C + 1$ classes. This means that your prediction should be 'none of these classes' if your prediction is not in one of { *ID classes* }.
Strictly follow the guidelines above.

Here are two examples of your response. The first one is when a given image is predicted into the given $C$ classes { *ID classes* }. The second one is when a given image is predicted into 'none of these classes'.

Please respond with the following examples format:

Example 1) when a given image is predicted into the given $C$ classes { *ID classes* }
--------------------------------------------------
Prediction: *example prediction*
Confidence: { *ID classes*: *example confidence* }
--------------------------------------------------

Example 2) when a given image is predicted into 'none of these classes'
--------------------------------------------------
Prediction: none of these classes
Confidence: { *ID classes*: *example confidence*, none of these classes: *example confidence* }
--------------------------------------------------

These above examples are confidences for two different images.
When you provide your answer, your answer format should be the same as the format between the dashed lines in the examples.
Strictly adhere to the examples format provided above. Do not deviate from the above format.

---

Figure B.6.1: Prompt used for experiments in Sec. 3.5

## B.6.2  PROMPT FOR REASONING

Fig. B.6.2 illustrates the prompt used for reasoning in Sec. 3.6. The red text indicates that the contents vary depending on the choice of benchmark. The text highlighted in yellow corresponds to the parts added for reasoning in the baseline prompt.

Your task is to classify the given image into one of these $C + 1$ classes: { *ID classes*, none of these classes } and assign confidence values for your prediction to each class , and give your rationale behind your prediction and confidence values.

You can classify the given image into 'none of these classes': if you cannot classify the given image into any of the given other $C$ classes, if you are not sure whether an image belongs to one of the given other $C$ classes, or if you think you need other classes other than the given other $C$ classes to classify the given image.

The following are guidelines for your response. Please respond according to these guidelines.
  - You should provide your confidence for each class between 0.00 and 100.00.
  - The confidence value can be expressed as a float.
  - The class you've assigned the max confidence should be your prediction.
  - Your confidence should be above 0.0 in at least one of the $C + 1$ classes given. It is not allowed to assign 0.0 confidence to every class.
  - You should provide your confidence in the $C + 1$ classes given, even if your prediction is 'none of these classes'.
  - Your prediction should be in the given $C + 1$ classes. You are not allowed to predict the given image into any other classes than the $C + 1$ given classes.
  - There is 'none of these classes' in the given $C + 1$ classes. This means that your prediction should be 'none of these classes' if your prediction is not in one of { *ID classes* }.
Strictly follow the guidelines above.

Here are two examples of your response. The first one is when a given image is predicted into the given $C$ classes { *ID classes* }. The second one is when a given image is predicted into 'none of these classes'.

Please respond with the following examples format:

Example 1) when a given image is predicted into the given $C$ classes { *ID classes* }
--------------------------------------------------
Prediction: *example prediction*
Confidence: { *ID classes: example confidence* }
Reasoning: *example reasoning*
--------------------------------------------------

Example 2) when a given image is predicted into 'none of these classes'
--------------------------------------------------
Prediction: none of these classes
Confidence: { *ID classes: example confidence*, none of these classes: *example confidence* }
Reasoning: *example reasoning*
--------------------------------------------------

These above examples are confidences for two different images.
When you provide your answer, your answer format should be the same as the format between the dashed lines in the examples.
Strictly adhere to the examples format provided above. Do not deviate from the above format.

Figure B.6.2: Prompt used for reasoning in Sec. 3.6

### B.6.3 PROMPT FOR CLASS ORDER

Fig. B.6.3 illustrates the prompt used for grouping classes in the ImageNet200 dataset based on their conceptual similarity.

I have 200 category names, and my goal is to group these categories into broader concepts for easier management. Here are the 200 category names I have.

Goldfish, great white shark, hammerhead, stingray, hen, ostrich, goldfinch, junco, bald eagle, vulture, newt, axolotl, tree frog, iguana, African chameleon, cobra, scorpion, tarantula, centipede, peacock, lorikeet, hummingbird, toucan, duck, goose, black swan, koala, jellyfish, snail, lobster, hermit crab, flamingo, american egret, pelican, king penguin, grey whale, killer whale, sea lion, chihuahua, shih tzu, afghan hound, basset hound, beagle, bloodhound, italian greyhound, whippet, weimaraner, yorkshire terrier, boston terrier, scottish terrier, west highland white terrier, golden retriever, labrador retriever, cocker spaniels, collie, border collie, rottweiler, german shepherd dog, boxer, french bulldog, saint bernard, husky, dalmatian, pug, pomeranian, chow chow, pembroke welsh corgi, toy poodle, standard poodle, timber wolf, hyena, red fox, tabby cat, leopard, snow leopard, lion, tiger, cheetah, polar bear, meerkat, ladybug, fly, bee, ant, grasshopper, cockroach, mantis, dragonfly, monarch butterfly, starfish, wood rabbit, porcupine, fox squirrel, beaver, guinea pig, zebra, pig, hippopotamus, bison, gazelle, llama, skunk, badger, orangutan, gorilla, chimpanzee, gibbon, baboon, panda, eel, clown fish, puffer fish, accordion, ambulance, assault rifle, backpack, barn, wheelbarrow, basketball, bathtub, lighthouse, beer glass, binoculars, birdhouse, bow tie, broom, bucket, cauldron, candle, cannon, canoe, carousel, castle, mobile phone, cowboy hat, electric guitar, fire engine, flute, gasmask, grand piano, guillotine, hammer, harmonica, harp, hatchet, jeep, joystick, lab coat, lawn mower, lipstick, mailbox, missile, mitten, parachute, pickup truck, pirate ship, revolver, rugby ball, sandal, saxophone, school bus, schooner, shield, soccer ball, space shuttle, spider web, steam locomotive, scarf, submarine, tank, tennis ball, tractor, trombone, vase, violin, military aircraft, wine bottle, ice cream, bagel, pretzel, cheeseburger, hotdog, cabbage, broccoli, cucumber, bell pepper, mushroom, Granny Smith, strawberry, lemon, pineapple, banana, pomegranate, pizza, burrito, espresso, volcano, baseball player, scuba diver, acorn

Here is the initial grouping I attempted.

- Goldfish, great white shark, hammerhead, stingray, grey whale, killer whale, sea lion,
- hen, ostrich, goldfinch, junco, bald eagle, vulture, flamingo, american egret, pelican, king penguin,
- jellyfish, snail, newt, axolotl, tree frog, iguana, African chameleon, cobra,
- eel, clown fish, puffer fish, starfish,
- lobster, hermit crab, scorpion, tarantula, centipede,
- peacock, lorikeet, hummingbird, toucan, duck, goose, black swan,
- chihuahua, shih tzu, afghan hound, basset hound, beagle, bloodhound, italian greyhound, whippet, weimaraner, yorkshire terrier, boston terrier, scottish terrier, west highland white terrier, golden retriever, labrador retriever, cocker spaniels, collie, border collie, rottweiler, german shepherd dog, boxer, french bulldog, saint bernard, husky, dalmatian, pug, pomeranian, chow chow, pembroke welsh corgi, toy poodle, standard poodle,
- timber wolf, hyena, red fox, tabby cat, leopard, snow leopard, lion, tiger, cheetah, meerkat,
- ladybug, fly, bee, ant, grasshopper, cockroach, mantis, dragonfly, monarch butterfly,
- wood rabbit, porcupine, fox squirrel, beaver, guinea pig, zebra, pig, hippopotamus, bison, gazelle, llama, skunk, badger, polar bear,
- orangutan, gorilla, chimpanzee, gibbon, baboon, panda, koala,
- ambulance, jeep, canoe, school bus, pickup truck, pirate ship, space shuttle, submarine, tractor, parachute,
- assault rifle, cannon, missile, revolver, tank, military aircraft,
backpack, barn, wheelbarrow, bathtub, lighthouse, beer glass, binoculars, birdhouse, bow tie, broom, bucket, cauldron, candle, carousel, castle, mobile phone, cowboy hat, fire engine, gasmask, guillotine, hammer,
basketball, rugby ball, soccer ball, tennis ball,
accordion, electric guitar, flute, grand piano, harmonica, harp, saxophone, trombone, violin,
hatchet,  joystick, lab coat, lawn mower, lipstick, mailbox, mitten, sandal, schooner, shield, spider web, steam locomotive, scarf, vase, wine bottle,
- ice cream, bagel, pretzel, cheeseburger, hotdog, pizza, burrito, espresso
- cabbage, broccoli, cucumber, bell pepper, mushroom,
- Granny Smith, strawberry, lemon, pineapple, banana, pomegranate, acorn
- volcano
- baseball player, scuba diver

Could you refine the grouping of these 200 classes to make them clearer?

Figure B.6.3: Prompt used for GPT-4o to group classes for class order in Sec. 3.6

## C    Experimental Details for Reflexive Guide

### C.1    Suggested Class Post-processing

Despite the guidelines provided in Stage 1, responses that deviated from the guidelines are observed. These failure cases can be broadly categorized into four cases: 1) inclusion of ID class names, 2) failure to suggest exactly $N$ classes for each near- and far-OoD group, 3) duplicated classes, and 4) no suggestion due to failure in recognizing the image. We post-process the suggested answers to refine them into a feasible set of auxiliary OoD classes, except for Case 4. For Case 1, we exclude suggestions that are character-level identical to ID classes. For Case 2, if the total number of suggestions exceeds $2N$, we sample only $2N$ classes from the suggestions. If the total number of suggestions is less than $2N$, we use the suggestions as is without any supplementation. For Case 3, we remove one of duplicated classes.

### C.2    Detailed Results for Sec. 4.1

Tab. C.2.1 presents the comprehensive results for ReGuide.

Table C.2.1: ReGuide results on $5\%$ subset of the ImageNet200 benchmark. The AURC values are multiplied by $10^3$, and the other values are percentages. 'Valid' indicates the ratio of valid responses out of a total of $4,170$ image-prompt queries. **Bold** denotes the best performance among the results from each model.

| Models | Valid | ID ImageNet200 ACC (↑) | ECE (↓) | AURC (↓) | Near-OoD NINCO | SSB-Hard | Far-OoD iNaturalist | Textures | Openimage-O | All OoD |
|---|---|---|---|---|---|---|---|---|---|---|
| | | | | | FPR@90%TPR (↓) / FPR@95%TPR (↓) / AUROC (↑) | | | | | |
| **InternVL2-26B** | 61.01 (2,544) | 91.23 | **8.12** | 92.27 | 82.73 / 82.73 / 58.31 | 94.34 / 94.34 / 52.51 | 38.03 / 38.03 / 80.66 | 28.86 / 28.86 / 85.25 | 47.91 / 47.91 / 75.72 | 73.12 / 73.12 / 63.11 |
| + GPT-text | 69.88 (2,914) | 89.58 | 11.74 | 82.48 | 69.44 / 69.44 / 62.17 | 85.65 / 85.73 / 53.55 | 26.82 / 28.00 / 84.72 | 29.20 / 29.20 / 83.51 | 39.39 / 39.39 / 78.10 | 62.41 / 62.64 / 65.88 |
| + ReGuide | 86.14 (3,592) | 93.53 | 11.94 | **55.63** | **22.39 / 22.89 / 86.53** | **15.21 / 15.21 / 90.41** | **1.39 / 1.39 / 98.02** | **3.93 / 3.93 / 97.05** | **2.04 / 2.04 / 97.68** | **10.24 / 10.27 / 93.19** |
| **InternVL2-76B** | 97.26 (4,056) | 89.09 | 5.97 | 79.87 | 51.28 / **51.28** / 71.89 | 71.02 / 71.02 / 62.02 | 2.20 / **2.20** / 96.43 | 10.76 / **10.76** / 92.15 | 14.27 / **14.27** / 90.40 | 44.46 / **44.46** / 75.30 |
| +ReGuide | 95.80 (3,995) | 90.93 | **1.54** | **9.28** | **8.05** / 56.36 / **91.35** | **14.58** / 66.65 / **87.65** | **0.00** / 59.75 / 95.35 | **4.08** / 60.00 / **93.38** | **2.02** / 65.46 / **93.95** | **8.92** / 64.36 / **90.60** |
| **GPT-4o** | 87.58 (3,652) | 90.64 | 2.25 | 10.14 | 8.57 / **14.76** / 93.96 | 29.25 / 34.50 / 82.28 | 0.81 / **1.83** / 98.11 | 5.60 / **6.47** / 95.37 | 1.21 / **3.63** / 97.82 | 15.62 / **19.34** / 89.85 |
| + ReGuide | 79.57 (3,318) | 91.59 | **1.54** | **8.57** | **0.49** / 18.72 / **96.76** | **7.53** / 31.17 / **92.56** | **0.00** / 17.05 / 97.08 | **1.32** / 26.43 / **95.96** | **0.15** / 19.66 / 96.82 | **4.02** / 25.66 / **94.61** |

* GPT-4o (2024-08-06), InternVL2-InternLM2-Chat-26B, InternVL2-LLaMA3-76B

As in Sec. B.2, we evaluate the effectiveness of ReGuide on the shared valid query set. As shown in Tab. C.2.2, the overall performance trends among the LVLMs remained consistent with their performance on their own valid query sets. ReGuide enhances OoDD performance across all models, consistent with the results observed on their own valid query sets. We additionally evaluate their FPR at a TPR threshold of $90\%$, considering the high sensitivity to TPR thresholds, as discussed in Sec. 3.6. At the $90\%$ TPR threshold, FPR is significantly reduced, and ReGuide consistently demonstrates lower FPR compared to the baseline across all OoD datasets.

Table C.2.2: ReGuide results on the shared valid query set of the 5% ImageNet200 benchmark. ↓ and ↑ mean that lower and higher values are better, respectively. The AURC values are multiplied by $10^3$, and the other values are percentages. **Bold** represents the best performance among the results from each model.

| Models | Valid | ID ImageNet200 ACC (↑) | ECE (↓) | AURC (↓) | Near-OoD NINCO | SSB-Hard | Far-OoD iNaturalist | Textures | Openimage-O | All OoD |
|---|---|---|---|---|---|---|---|---|---|---|
| | | | | | FPR@90%TPR (↓) / FPR@95%TPR (↓) / AUROC (↑) | | | | | |
| **InternVL2-26B** | | 93.30 | **6.15** | 61.82 | 86.67 / 86.67 / 56.39 | 93.78 / 93.78 / 52.83 | 34.03 / 34.03 / 82.70 | 31.52 / 31.52 / 83.96 | 47.20 / 47.20 / 76.12 | 70.34 / 70.34 / 64.55 |
| + ReGuide | | 96.09 | 9.22 | 39.58 | 33.33 / 33.33 / 81.75 | 17.73 / 17.88 / 90.21 | 2.09 / 2.09 / 98.75 | 5.43 / 6.52 / 96.91 | 2.00 / 2.00 / 98.84 | 12.23 / 12.39 / 93.22 |
| **InternVL2-76B** | 34.29 | 92.74 | 5.03 | 71.12 | 65.33 / 65.33 / 66.22 | 76.98 / 76.98 / 60.39 | 1.05 / **1.05** / 98.36 | 10.87 / **10.87** / 93.45 | 18.40 / **18.40** / 89.68 | 48.12 / **48.12** / 74.82 |
| +ReGuide | (1,430) | 93.85 | **1.40** | **9.17** | **6.67 / 56.00 / 93.27** | **19.75 / 73.09 / 86.39** | **0.00** / 64.92 / 96.46 | **6.52** / 69.57 / 93.14 | **2.80** / 60.40 / **95.10** | **11.59** / 68.03 / **90.58** |
| **GPT-4o** | | 93.85 | 1.82 | **2.92** | 8.00 / **9.33** / 93.86 | 32.81 / **34.68** / 80.98 | 1.05 / **1.05** / 97.92 | 8.70 / **8.70** / 93.66 | 1.60 / **1.60** / 97.73 | 18.47 / **19.50** / 88.62 |
| + ReGuide | | 92.18 | **1.39** | 11.22 | **1.33** / 28.00 / **96.42** | **9.80** / 42.77 / **91.41** | **0.00** / 25.13 / 97.13 | **3.26** / 33.70 / **95.12** | **0.40** / 33.20 / 96.62 | **5.44** / 36.61 / **93.90** |

* GPT-4o (2024-08-06), InternVL2-InternLM2-Chat-26B, InternVL2-LLaMA3-76B

Tab. C.2.3 and C.2.4 present example responses of each models in Tab. 4. The first row in Tab. C.2.3 indicates the classes suggested by GPT-4o when ID class names are provided (i.e., GPT-text). The prompt used to obtain class suggestions of GPT-text can be found in Sec. C.5. A `class name` below each image is the ground truth of the corresponding image. The class suggestion set in GPT-text remains fixed regardless of the input images, as ID classes are static information, whereas ReGuide's suggestions change based on the input image due to its image-adaptive nature.

The results on InternVL2-26B/-76B and GPT-4o demonstrate the effectiveness of ReGuide. Firstly, ReGuide suggests helpful classes for OoDD. For example, as shown in Tab. C.2.3, In-

ternVL2+ReGuide suggests near-OoD classes for `dotted` image, such as `polka dot`, `cherry red`, or `salmon-colored`, which are highly close to its ground-truth. These suggested classes successfully guide the model to classify OoD inputs into one of the auxiliary OoD classes. For `softball` image, the prediction of ReGuide is also closest to the ground-truth.

Table C.2.3: Example responses of InternVL2-26B on 5% subset of ImageNet200 benchmark

| | Near-OoD | Far-OoD | Prediction |
|---|---|---|---|
| | **GPT-text suggested OoD classes** | | InternVL2-26B
+ GPT-text
+ ReGuide |
| | siamese cat, bengal tiger cub, cheetah cub, lynx, arctic wolf, manatee, walrus, otter, arctic fox, albatross, macaw, snowy owl, peregrine falcon, arctic hare, hedgehog, woodpecker, puffin, seahorse, swordfish, caracal | motorcycle, bulldozer, windmill, skyscraper, piano keyboard, surfboard, basketball hoop, bookshelf, drone, traffic light, skis, space rover, diving board, ferris wheel, telescope, typewriter, electric fan, hot air balloon, sewing machine, xylophone | |
| | **ReGuide suggested OoD classes** | | |
| **ImageNet200**  hammerhead | marlin, bonito, remora, great white shark, sailfish, tiger shark, swordtail, flying fish, barracouta, menhaden, electric eel, smelt, pilotfish, thresher shark, squid, tuna, herring, bluefin tuna, mackerel, pufferfish | nautilus, sea urchin, sand dollar, starfish, oyster, krill, cuttlefish, sea spider, snail, horseshoe crab, mahimahi, conch, guppy, copepod, abalone, sunfish, barracuda, octopus, brittle star, mullet | great white shark
swordfish
hammerhead |
| **SSB Hard**  softball | baseball diamond, baseball uniform, baseball field, baseball helmet, umpire, athlete, baseball game, softball player, baseball team, baseball, outfielder, baseball bat, baseball jersey, second baseman, pitcher | baseball pants, fielder, catcher, shortstop, baseball stadium, sports player, first baseman, baseball glove, baseball cap, baseball card, third baseman, coach, baseball mitt, baseball player, baseball cleats, batter, referee | baseball player
baseball player
softball player |
| **Textures**  dotted | polka dot, cerise, cherry red, blush, cherry blossom, fuchsia, garnet, coral, rose, peach, roseate, salmon-colored | cherry, pink, vermilion, crimson, wine red, salmonpink, scarlet, magenta, ruby, maroon, burgundy, salmon pink, mauve | scarf
none of these classes
polka dot |

Table C.2.4: Example responses of GPT-4o on 5% subset of ImageNet200 benchmark

| | | ReGuide suggested OoD classes | | Prediction GPT-4o + ReGuide |
|---|---|---|---|---|
| | | **Near-OoD** | **Far-OoD** | |
| **ImageNet200** | whippet | greyhound, lurcher, saluki, rhodesian ridgeback, vizsla, basenji, irish wolfhound, doberman, ibizan hound, deerhound, basque shepherd, great dane, galgo, pharaoh hound, borzoi | umbrella, wristwatch, bicycle, electric fan, soccer cleat, refrigerator, telescope, skateboard, basketball hoop, vacuum cleaner, washing machine, piano, coffee maker, laptop, microwave, tennis racket, desk chair, bookcase, alarm clock | greyhound / whippet |
| **SSB Hard** | yellowhammer | oriole, bunting, grosbeak, vireo, sparrow, kinglet, finch, linnet, wren, pine siskin, nuthatch, phoebe, treecreeper, tanager, canary, starling, chickadee, warbler, titmouse | umbrella, wristwatch, bicycle, electric fan, soccer cleat, alarm clock, tennis racket, refrigerator, telescope, skateboard, basketball hoop, vacuum cleaner, washing machine, piano, laptop, microwave, desk chair, bookcase, coffee maker | goldfinch / none of these classes |
| **Textures** | scaly | fish scales, coral, chainmail, turtle shell, reptile skin, armadillo shell, rocky surface, crocodile skin, bark texture, mesh fabric, mineral formation, mosaic, sandstone, fossil texture, lizard skin, dragon scales, pangolin scales, pebbles, snake skin, textured leather | umbrella, wristwatch, bicycle, electric fan, soccer cleat, alarm clock, tennis racket, refrigerator, telescope, skateboard, basketball hoop, vacuum cleaner, washing machine, piano, laptop, microwave, desk chair, bookcase, coffee maker | goldfish / fish scales |

## C.3 ANALYSIS ON OoD SCORE WITH REGUIDE

To further investigate the FPR increase in larger models such as InternVL2-76B and GPT-4o in Tab. 4, we analyze the ROC curves for each model with and without ReGuide. Fig. C.3.1 shows the ROC curves for InternVL2-26B and GPT-4o. Both models exhibit improved ROC curves after applying ReGuide; however, the FPR declines more gradually for GPT-4o+ReGuide compared to InternVL2-26B+ReGuide as TPR decreases from 100% as shown in Fig. C.3.1(b) and C.3.1(d). While the reduction in FPR for GPT-4o+ReGuide is less pronounced at high TPR thresholds (e.g., 100%–95%), it achieves significantly lower FPR as the TPR threshold approaches 90%.

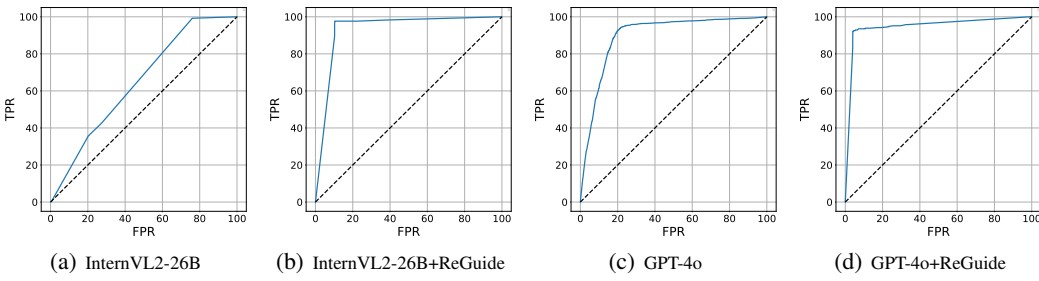

| (a) InternVL2-26B | (b) InternVL2-26B+ReGuide | (c) GPT-4o | (d) GPT-4o+ReGuide |

Figure C.3.1: ROC curves for InternVL2-26B and GPT-4o

To explore the causes of the FPR change observed across models in Fig. C.3.1, we analyze examples at the two extremes –100.00% and 0.0%– of the ROC curves, as well as the shifts in OoD score distributions induced by ReGuide. This analysis reveals that the increase in FPR@95%TPR for larger models such as InternVL2-76B and GPT-4o stems from their superior visual recognition ability, which leads to more confident predictions with highly suitable class suggestions.

Tab. C.3.1 presents examples on the extremes of Fig. C.3.1(b) and C.3.1(d). This highlights that labeling issues arise from insufficiently thorough annotations. First, a small subset of datasets exhibits a conceptual mismatch between the objects in images and their ground-truth labels. For instance, the `grand piano` image demonstrates a misprediction caused by such labeling issues, as the object is

more accurately described as a `grand piano mechanism`, a class suggested by GPT-4o, rather than the ground-truth label `grand piano`. Another case involves images with ground-truth labels corresponding to objects that occupy only a small portion of the image, as shown in the `bee` and `ball` examples. In the `bee` image, the dominant object is a plant resembling a `cactus`, yet the ground-truth label is `bee`. For such images, LVLMs often suggest classes based on the dominant objects and classify them accordingly.

Table C.3.1: Examples at the extremes of the ROC curves for the ImageNet200 benchmark. **Bold** indicates the predicted class and its confidence score before applying the softmax function. Classes with a confidence score of $0.00$ have been omitted.

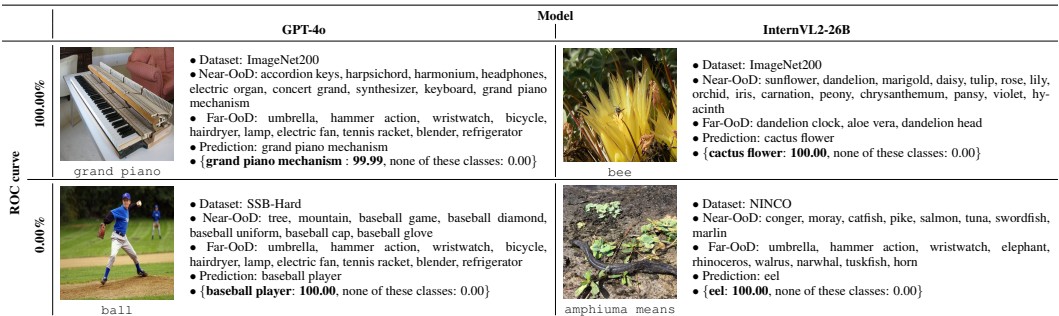

The suggested classes provided by GPT-4o for label-mismatched inputs are significantly fine-grained and contextually appropriate. Tab. C.3.2 presents the classes suggested by GPT-4o for images with ambiguous or inaccurate labels. For example, GPT-4o suggests classes such as `model ape`, `gorilla figurine`, `orangutan statue`, and `monkey figurine` as near-OoD classes. Based on the shape and texture of the object in the image, these suggestions are highly relevant and more suitable classifications. Similarly, for the `cockshell` image, GPT-4o suggests near-OoD classes such as `motorboat`, `sailboat`, `rowboat`, and `kayak`, which are fine-grained categories closely related to the depicted object. As shown in Tab. C.3.2, GPT-4o's strong image recognition capabilities allow it to provide fine-grained, contextually appropriate classifications for images with ambiguous or inaccurate labels.

Table C.3.2: GPT-4o+ReGuide examples for images with ambiguous or inaccurate labels from the ImageNet200 benchmark. **Bold** indicates the predicted class and its confidence score before applying the softmax function. Classes with a confidence score of $0.00$ have been omitted.

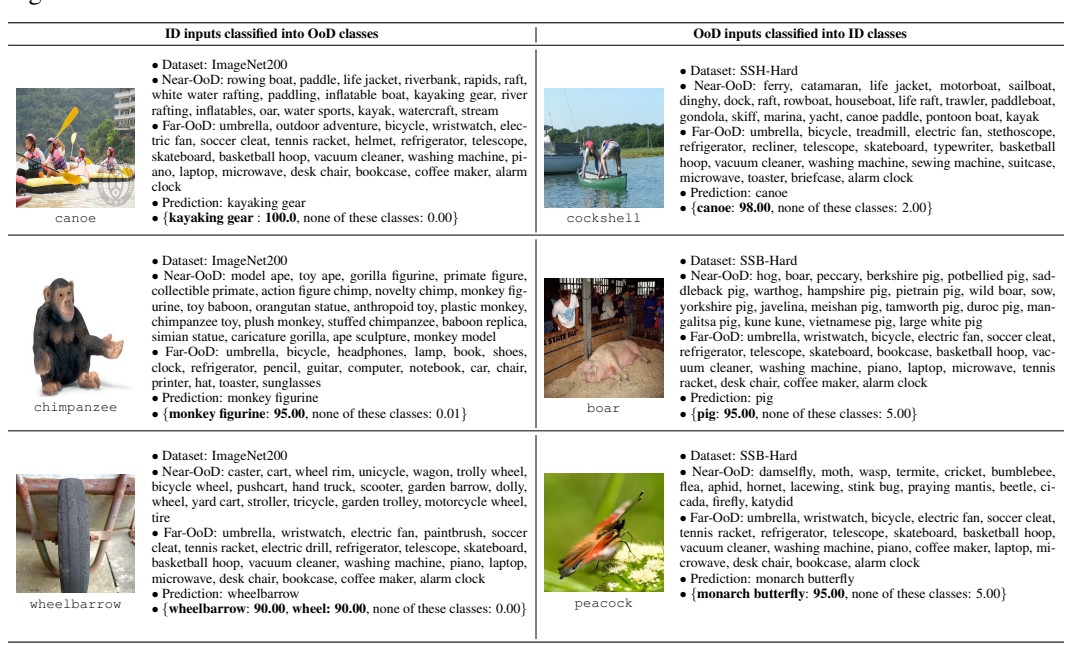

Fig. C.3.2(a) and C.3.2(b) depict the OoD score distributions for InternVL2-26B and GPT-4o, respectively. As shown in Fig. C.3.2(a), with ReGuide, InternVL2-26B produces OoD scores ranging between 0.00 and 100.00, which is not observed without ReGuide. However, the effect of ReGuide on GPT-4o differs from that on InternVL2-26B. The OoD score distribution for GPT-4o becomes more biased with ReGuide, as illustrated in Fig. C.3.2(b). The number of responses with OoD scores between 0.00 and 100.00 decrease significantly. We infer that this outcome is due to the strong visual recognition ability of GPT-4o. As shown in Tab. 2, the suggestions of GPT-4o include classes that are highly similar to the given image.

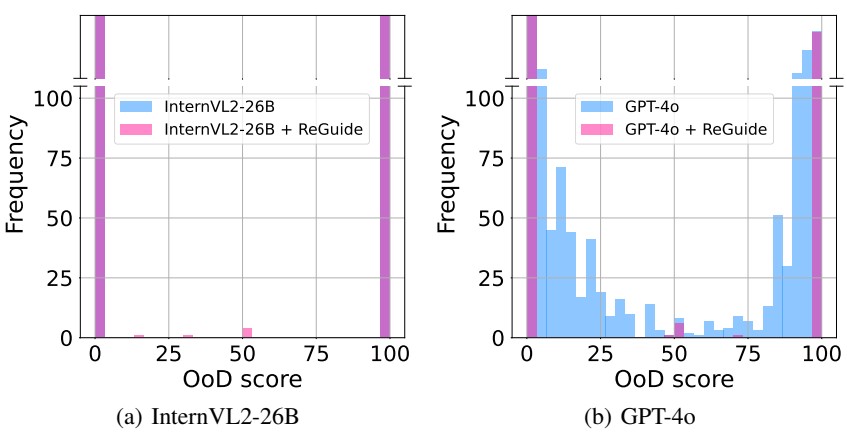

(a) InternVL2-26B          (b) GPT-4o

Figure C.3.2: OoD score distributions for InternVL2-26B and GPT-4o

For more detailed analysis of the ReGuide effect, we examine the OoD score distributions for ID and OoD inputs separately. Fig. C.3.3 displays OoD score distributions for ID and OoD inputs on InternVL2-26B and GPT-4o.

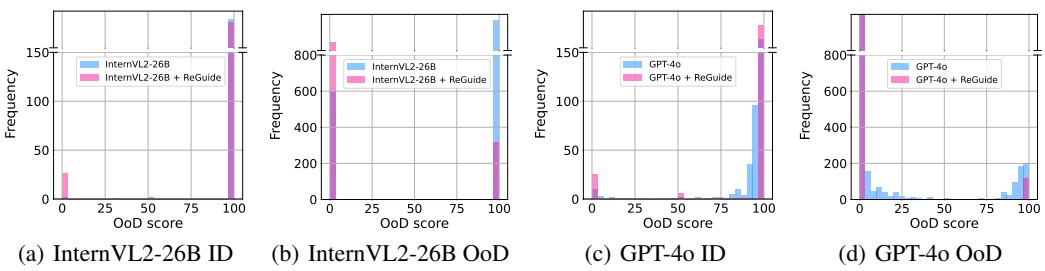

(a) InternVL2-26B ID    (b) InternVL2-26B OoD    (c) GPT-4o ID    (d) GPT-4o OoD

Figure C.3.3: OoD scores distributions of each ID and OoD for InternVL2-26B and GPT-4o

A common observation regarding ReGuide for InternVL2-26B and GPT-4o is the shift in the OoD score distribution of OoD inputs. As shown in Fig. C.3.3(b) and C.3.3(d), ReGuide increases the number of OoD inputs with an OoD score of $0.0$ while reducing those with a score of $100$. This shift demonstrates that the models effectively classify OoD inputs into one of the suggested OoD classes, thereby improving OoD detectability and confirming that ReGuide functions as intended.

The increase in inputs with an OoD score of $0.0$ is also observed for ID inputs. As previously discussed in Tab. C.3.1, this is caused by ID images with ambiguous or inaccurate labels. For such inputs, GPT-4o+ReGuide effectively classifies them into a single class with higher confidence, as the suggested classes often include one closely aligned with the ground-truth label. This explanation is further supported by the improvement in FPR@90%TPR, where the influence of this small subset of images is reduced, as shown in Tab. 4.

## C.4 COMPARISON OF OoD SCORE FUNCTIONS

When incorporating available auxiliary OoD classes, one of effective scoring strategies is to include confidence estimates for these auxiliary OoD classes. For example, NegLabel (Jiang et al., 2024) uses the scoring function as the ratio of the sum of ID softmax to the sum of both ID and OoD softmax, and EOE (Cao et al., 2024) utilizes the difference between max ID softmax and max OoD softmax.

We compare three different scoring functions: 1) the maximum ID softmax, 2) the difference between the sum of ID softmax and OoD softmax, and 3) the difference between the maximum ID softmax and maximum OoD softmax. Each of the scoring functions is represented, in order, as follows:

$$\mathcal{S}_{\text{max}}(\mathbf{x}, \mathcal{Y}, \mathcal{A}_{\text{aux}}) = \max_{i \in \mathcal{Y}} \frac{e^{s_i/\tau}}{\sum_{k \in \mathcal{Y} \cup \mathcal{A}_{\text{aux}}} e^{s_k/\tau}} \tag{1}$$

$$\mathcal{S}_{\text{sumsub}}(\mathbf{x}, \mathcal{Y}, \mathcal{A}_{\text{aux}}) = \sum_{i \in \mathcal{Y}} \frac{e^{s_i/\tau}}{\sum_{k \in \mathcal{Y} \cup \mathcal{A}_{\text{aux}}} e^{s_k/\tau}} - \sum_{j \in \mathcal{A}_{\text{aux}}} \frac{e^{s_j/\tau}}{\sum_{k \in \mathcal{Y} \cup \mathcal{A}_{\text{aux}}} e^{s_k/\tau}} \tag{2}$$

$$\mathcal{S}_{\text{maxsub}}(\mathbf{x}, \mathcal{Y}, \mathcal{A}_{\text{aux}}) = \max_{i \in \mathcal{Y}} \frac{e^{s_i/\tau}}{\sum_{k \in \mathcal{Y} \cup \mathcal{A}_{\text{aux}}} e^{s_k/\tau}} - \max_{j \in \mathcal{A}_{\text{aux}}} \frac{e^{s_j/\tau}}{\sum_{k \in \mathcal{Y} \cup \mathcal{A}_{\text{aux}}} e^{s_k/\tau}} \tag{3}$$

where $\tau$ is a temperature.

Tab. C.4.1 presents the results with these scoring functions. All results are on a $5\%$ subset of ImageNet200 benchmark using InternVL2-26B+ReGuide. No significant differences are observed across the scoring functions. LVLMs tend to assign high confidence to a single predicted class, resulting in similar results regardless of these different scoring designs. Therefore, all results reported in our paper are measured by $\mathcal{S}_{\text{max}}$.

Table C.4.1: Comparison results between different scoring functions on a $5\%$ subset of ImageNet200 benchmark using InternVL2-26B

| | Near-OoD | | Far-OoD | | | All OoD |
|---|---|---|---|---|---|---|
| | NINCO | SSB-Hard | iNaturalist | Textures | Openimage-O | |
| | FPR@95%TPR ($\downarrow$) / AUROC ($\uparrow$) | | | | | |
| $\mathcal{S}_{\text{max}}$ | 22.89 / 86.53 | 15.21 / 90.41 | 1.39 / 98.02 | 3.93 / 97.05 | 2.04 / 97.68 | 10.27 / 93.19 |
| $\mathcal{S}_{\text{sumsub}}$ | 23.88 / 86.46 | 15.21 / 90.58 | 10.65 / 97.45 | 3.49 / 96.90 | 2.33 / 97.69 | 11.59 / 93.19 |
| $\mathcal{S}_{\text{maxsub}}$ | 23.38 / 86.53 | 15.27 / 90.59 | 1.39 / 97.45 | 3.93 / 96.90 | 2.04 / 97.69 | 10.33 / 93.26 |

## C.5 PROMPT FORMAT

We provide the full prompts used for experiments in Sec. 4.1. In the prompt, yellow, green, red, blue boxes indicate a task description, an explanation of rejection class, guidelines, and examples for response format, respectively.

### C.5.1 PROMPT FOR CLASS SUGGESTION OF GPT-TEXT

Fig. C.5.1 illustrates the prompt used for class suggestion of GPT-text. The guidelines consist of two key instructions: first, to avoid suggesting classes that are too similar to the image, and second, to avoid suggesting classes that are identical to the ID class. The red text indicates that the contents vary depending on the choice of benchmark.

---

Your task is to provide 2N class names based on the given class names: {*ID classes*}. The first N names are visually similar to the given classes. The last N names are visually dissimilar or belong to different domains. You should not provide over 2N class names.

Avoid suggestions that are very similar to each other. For example, suggestions that are part of a large object or in the same broad category, such as {*example class set* 1} or {*example class set* 2}.

Do not provide the following category names as your suggestions: {*ID classes*, none of these classes}

---

Figure C.5.1: Prompt used for class suggestion of GPT-text

### C.5.2 PROMPT FOR REGUIDE STAGE 1

Fig. C.5.2 illustrates the prompt used for class suggestion of ReGuide Stage 1. Similar to the original prompt used for evaluating OoDD and the prompt used for class suggestion on GPT-text, we provide a few guidelines to prevent receiving uninformative class names, such as ID class names, overly similar class names or just a repetition of a class name. Without an example, the response format of image-based suggestions deviates more compared to that of text-based suggestions. Thus, we include the response format as an illustrative example. The red text indicates that the contents vary depending on the choice of benchmark.

---

Your task is to provide 2N class names based on the given image. The first N names are visually similar to the image. The last N names are visually dissimilar or belong to different domains. You should not provide over 2N class names.

Avoid suggestions that are very similar to each other. For example, suggestions that are part of a large object or in the same broad category, such as {*example class set* 1} or {*example class set* 2}.

Do not provide the following category names as your suggestions: {*ID classes*, none of these classes}

Here is an example of your response:
Example) When the given image is '*example label*'
------------------------------------------
Class suggested: *example class suggestions*
------------------------------------------

Strictly follow this format for your response.

---

Figure C.5.2: Prompt used for ReGuide Stage 1

### C.5.3 PROMPT FOR REGUIDE STAGE 2

Fig. C.5.3 illustrates the prompt used for ReGuide Stage 2. The red text indicates that the contents vary depending on the choice of benchmark. The blue indicates the locations of the suggested OoD classes from Stage 1. The highlighted text in yellow indicates modifications made to reflect these suggested classes. GPT-text employs the same prompt format for GPT-4o's suggested classes.

Your task is to classify the given image into one of these $C + 1$ classes: {ID classes, auxiliary OoD classes, none of these classes} and assign confidence values for your prediction to each class.

You can classify the given image into 'none of these classes': if you cannot classify the given image into any of the other classes given, if you are not sure whether an image belongs to one of the given other $C + 2N$ classes, or if you think you need other classes other than the given other $C + 2N$ classes to classify the given image.

The following are guidelines for your response. Please respond according to these guidelines.
- You should provide your confidence for each class between 0.00 and 100.00.
- The confidence value can be expressed as a float.
- The class you've assigned the max confidence should be your prediction.
- Your confidence should be above 0.0 in at least one of the $C + 2N + 1$ classes given. It is not allowed to assign 0.0 confidence to every class.
- You should provide your confidence in the $C + 2N + 1$ classes given, even if your prediction is 'none of these classes'.
- Your prediction should be in the given $C + 2N + 1$ classes. You are not allowed to predict the given image into any other classes than the $C + 2N + 1$ given classes.
- There is 'none of these classes' in the given $C + 2N + 1$ classes. This means that your prediction should be 'none of these classes' if your prediction is not in one of the other classes.
Strictly follow the guidelines above.

Here is an example of your response.

Please respond with the following examples format:

Example)
---------------------------------------------------
Prediction: your prediction
Confidence: {ID classes: your prediction, auxiliary OoD classes: your prediction, none of these classes: your prediction}
---------------------------------------------------

These above examples are confidences for two different images.
When you provide your answer, your answer format should be the same as the format between the dashed lines in the examples.
Strictly adhere to the examples format provided above. Do not deviate from the above format.

Figure C.5.3: Prompt used for ReGuide Stage 2

### C.6 INFERENCE COST ANALYSIS

While ReGuide, as a two-stage approach, incurs higher computational costs compared to single-turn generation (Section 3), the inference cost does not double. This is because caching image representations eliminates the need to process the same image twice, and the shorter prompt in Stage 1 (class suggestion) reduces the token count. For open-source models, we analyze the inference cost from a computational complexity perspective, whereas for proprietary models, the analysis focuses on API usage costs.

We estimate the inference time $FLOPs$ following Li et al. (2024b). This standard practical estimate is based on a simplified scaling law, assuming that the inference cost of the LLM scales linearly with both the number of tokens processed and the model size. The inference time $FLOPs$ is estimated as:

$$FLOPs = \mathcal{O}(N \times T) \tag{4}$$

where $N$ is the number of parameters in the LLM and $T$ denotes the total number of tokens involved in the inference process. Specifically, $T$ is the sum of $Q$, $V$, and $G$, which correspond to the number of text input tokens, image tokens, and generated tokens, respectively. For example, for InternVL2-76B, the prompt depicted in Fig. B.6.1 with ImageNet200 classes includes 5,350 input tokens. The number of generated tokens is 1,936, assuming the output follows the provided example structures, and the number of image tokens is 256. Thus, the baseline $FLOPs$ is estimated as $7 \times 10^{10} \times (256 + 5,350 + 1,936)$.

Similarly, for ReGuide, $FLOPs$ is estimated as:

$$FLOPs = \mathcal{O}(N \times (V + Q_{Stage1} + G_{Stage1} + Q_{Stage2} + G_{Stage2})) \tag{5}$$

where $Q_{Stage1}, G_{Stage1}$ and $Q_{Stage2}, G_{Stage2}$ denote the number of text input and generated tokens for Stage 1 and Stage 2, respectively. Although ReGuide involves two stages, the input image is processed only once as the same image is reused across both stages and can be cached. Note that the inference cost of the vision encoder is excluded, as it remains consistent for both the baseline and ReGuide within the same VLM.

Tab. C.6.1 presents $FLOPs$ for the baseline and ReGuide on InternVL2 models. These values are estimated on one input pair (image-prompt) of the ImageNet200 benchmark, assuming that the generated outputs follow one of the provided example structures.

ReGuide requires only $1.14\times$ the computational cost compared to the baseline. It is because 1) the prompt used in Stage 1 is short and simple (e.g., 917 tokens for Stage 1), and 2) the image tokens are processed only once. Consequently, the increase in computational cost arises solely from the processing of the Stage 1 prompt. Additional optimization techniques for multi-turn inference could further reduce the computational cost gap between the baseline and ReGuide.

Tab. C.6.2 shows API costs for proprietary models under the baseline and ReGuide experiments on the 5% ImageNet200 benchmark. From the perspective of API costs, ReGuide exhibited a higher proportional increase compared to computational costs above, with a $1.47\times$ increase relative to the baseline. Since we processed each stage separately, the API cost reflects a scenario where image token caching is not applied. Additionally, we did not employ multi-turn inference, as it provided no advantages in inference time due to the absence of batch inference support. Implementing multi-turn batch inference could significantly improve efficiency and reduce costs.

Table C.6.1: Estimated computational costs for InternVL2 models (in units of $10^{13}$ FLOPs)

| Models | Baseline | +ReGuide |
|---|---|---|
| **InternVL2-26B** | 15.086 | 17.052 |
| **InternVL2-76B** | 52.794 | 60.144 |

[*] InternVL2-InternLM2-Chat-26B, InternVL2-LLaMA3-76B

Table C.6.2: API costs (USD) for proprietary models, calculated per 1 million input and output tokens

| Models | Input token | Output token | Baseline | ReGuide |
|---|---|---|---|---|
| **GPT-4o** | 1.25 | 5 | 73 | 107 |
| **Claude 3.5 Sonnet** | 3 | 15 | 228 | - |
| **Gemini Pro 1.5** | 3.5 | 10.5 | 193 | - |

[*] GPT-4o (2024-08-06)

One possible approach to reduce the inference cost gap between the baseline and ReGuide is to integrate the two-stage prompts into a single prompt while maintaining ReGuide's process of providing suggested classes. This adjustment would allow ReGuide to maintain its effectiveness with a single forward pass. Fundamentally, it is essential to develop LVLMs to overcome their tendency to limit responses to the options explicitly provided in the user-defined prompt. Such advancements would enable LVLMs to explore and utilize information beyond the immediate constraints of the prompt, establishing robust models capable of generating safe and reliable responses to a wider range of inputs.

## C.7 LIMITATIONS OF REGUIDE

While the proposed ReGuide improves performance in both image classification and OoDD tasks, there remain limitations. These challenges are primarily caused by the difficulty in exerting precise control over the behavior of the VLM. Due to the inherent complexity of understanding the implicit dynamics of LVLMs, controlling their behavior through prompting does not always ensure the desired outcome. Although we provide additional guidelines based on the model's failure cases to mitigate unintended responses, this approach is insufficient for full control over outputs. To address unintended outputs, we can utilize LLMs to iteratively refine prompts based on the model's responses. Leveraging external LLMs for this purpose is expected to function similarly to model ensemble principles, resulting in prompts better suited to the model compared to heuristic refinements.

From a visual perspective, ReGuide relies on the image understanding capabilities of LVLMs. Any potential improvement depends on the model's ability to recognize visuals accurately. Additionally, since our proposed approach is image-adaptive, the quality of the suggested class set is influenced by the image context. For instance, if the target object occupies only a small portion of the image, an LVLM may focus more on other objects in the background. This tendency can result in class suggestions that are either uninformative or overly broad.

