# OpenReview forum: "Reflexive Guidance: Improving OoDD in Vision-Language Models via Self-Guided Image-Adaptive Concept Generation"
_ICLR.cc/2025/Conference — ICLR 2025 Poster_

### Official Review · Reviewer_wyWF · 2024-10-28

**Soundness:** 2
**Presentation:** 2
**Contribution:** 2
**Rating:** 6
**Confidence:** 3

**Summary:**

This paper explores to enhance out-of-distribution detection (OoDD) in large vision-language models (LVLMs). The authors emphasize the increasing importance of these models in various domains and highlight the challenges related to their trustworthiness, specifically in dealing with out-of-distribution (OoD) inputs. To address this, they propose a self-guided prompting approach called Reflexive Guidance (ReGuide), which leverages LVLMs’ visual interpretation capabilities to improve OoD detection by generating image-adaptive concept suggestions. The experiments demonstrate that ReGuide significantly boosts the performance of both open-source and proprietary models in OoD detection tasks.

**Strengths:**

1. The introduction of the ReGuide method is interesting. By using image-adaptive concept generation, the authors provide a unique solution to enhance OoD detection in LVLMs. This approach leverages the strong visual recognition capabilities of these models, offering a novel way to bridge gaps in OoD detectability.

2. Comprehensive Evaluation: This paper comprehensively evaluates both proprietary and open-source LVLMs, providing a balanced comparison across different models. This may be helpful for future research.

3. Detailed Analysis: The authors provide a well-rounded analysis, including experimental results, visual feature interpretation, and the effectiveness of ReGuide on near- and far-OoD datasets. This depth of analysis gives credibility to their findings and demonstrates the robustness of their proposed method. Furthermore, it demonstrates the potential of introducing LVLMs to the area,

**Weaknesses:**

1. Limited Explanation of Failures: Although the paper provides examples of failure cases, it does not fully delve into the underlying causes or offer detailed solutions for these issues. While the authors acknowledge the problem of overconfidence in models such as GPT-4o with ReGuide, further exploration of these limitations could strengthen the study. There is still not a robust method to effectively introduce LVLMs to the OoD tasks.

2. Model-Specific Insights: The paper focuses on generic findings across models, but a deeper investigation into how specific models (e.g., GPT-4o vs. InternVL2) behave differently when ReGuide is applied could add nuance to the conclusions. For example, the differences in false positive rates (FPR) between models with and without ReGuide should be presented for a better comparison.

3. Scalability and Practicality: While the ReGuide method shows a promising direction, the computational overhead and API limitations mentioned in the paper could present challenges for practical, large-scale implementation. This issue is touched upon but not sufficiently addressed in terms of how ReGuide might be optimized for deployment at scale. Meanwhile, the inference cost analysis can further improve the paper's quality and inspire further work.

**Questions:**

N/A

---

> ### Author Response · Authors · 2024-11-22
> **Response to Reviewer wyWF (Part 1/3)**
>
> Thank you for your insightful comments and constructive feedback on our paper. Please find our point-to-point responses below, and refer to [Responses to all reviewers] for an overview of the revisions made to the manuscript. If you have any additional concerns or require further clarification, please do not hesitate to let us know. We would be happy to address any questions.
>
> ---
> ### **[Limited explanation of failures]**
>
> Our prior analysis of failure cases, presented in Appendix B.5, was aimed at investigating the patterns of failure cases specific to each LVLM. We agree that providing detailed solutions to mitigate such failures is important to understanding LVLM behavior and facilitating future research.
>
> It is important to note that the design of the prompt was **not intended to optimize OoDD performance but rather to reduce the number of failure cases** in LVLM responses. To achieve this, we included detailed guidelines and examples, and the effectiveness of this approach is demonstrated by the significant reduction in failure cases, as shown in the table below. **These results have been updated in Appendix B.5**.
>
> | InternVL2-76B      | (1) | (2) | (3) | (4) | (5) |  (6) | Total |
> |--------------------|-----:|-----:|-----:|-----:|-----:|-----:|-------:|
> | Baseline           | 193 | 886 |  -  |  0  | 201 |  0  | 1280  |
> |&nbsp;+ ID example       | 18  | 731 |  0  |  0  | 100 |  0  |  849  |
> |  &nbsp; &nbsp;+ Guidelines     | 10  | 493 |  0  |  0  | 128 |  0  |  631  |
> |   &nbsp; &nbsp; &nbsp;  + OoD example  | 26  | 34  |  0  |  0  | 79  |  8  |  147  |
>
> Baseline refers to a prompt that includes only the basic elements: a task description and an explanation of the rejection class. Adding guidelines and examples leads to a nearly 90% reduction in failure cases compared to when they are not included. Examples help address all types of failure cases, while guidelines are especially effective in reducing Failure Case (2), where responses do not follow the required format.
>
> ---
> ### **[Further exploration of limitations and model-specific insights]**
>
> In the original manuscript, AUROC and FPR@95%TPR were calculated using an overly strict approach that prioritized OoD inputs lower than ID inputs with the same score. This approach sometimes resulted in FPR@95%TPR values reaching 100%. However, we found that this method made the results harder to interpret and could lead to misunderstanding. To address this, we recalculated AUROC and FPR@95%TPR using a standard interpolation method commonly employed in libraries such as scikit-learn, and this has been updated to the revised manuscript.
>
> We would like to clarify that this adjustment does not affect the main findings of our study. However, it does lead to the following changes in the interpretation of FPR variations before and after applying ReGuide in Section 4:
> - Previously, InternVL2-76B and GPT-4o showed different trends in FPR@95%TPR. With the updated calculations, both models now show a similar pattern: FPR@95%TPR increases for both models after applying ReGuide.
> - This increase is caused by more ID samples receiving OoD scores of 0.0 after applying ReGuide. The primary reason is that some ID samples have ground-truth labels that either lack a strong correspondence to the object in the image or correspond to only one of multiple objects present in the image. Models with strong visual interpretation capabilities, such as InternVL2-76B and GPT-4o, generally suggest more appropriate classes for such ID inputs during Stage 1 of ReGuide. These models then assign high confidence scores to these suggested classes, which increases FPR@95%TPR.
>   - For example, as shown in the newly updated Table C.3.1, for an ID image with the ground-truth label "grand piano", GPT-4o suggested the class "grand piano mechanism", which can be considered more aligned with the image. In another case, where both a "bee" and a "cactus flower" are present but the ground-truth label is "bee",  InternVL2-26B suggested the more visually prominent class "cactus flower", resulting in an OoD score of 0.0. These examples demonstrate that LVLMs can suggest classes that better correspond to the image content and classify the images accordingly.
> - Importantly, this observation is not a negative effect of ReGuide but a result of a small subset of ID samples with ambiguous or inaccurate labels. This explanation is further supported by the improvement in FPR@90%TPR, which shows that ReGuide generally enhances the model’s ability to handle other ID and OoD inputs effectively as the influence of this small subset of images is diminished.
> **A detailed analysis, including the ROC curve, OoD score distribution, and examples of ID inputs with ambiguous and inaccurate labels can be found in Appendix C.3**.

---

> ### Author Response · Authors · 2024-11-22
> **Response to Reviewer wyWF (Part 2/3)**
>
> ---
> ### **[Scalability and Practicality] (1/2)**
>
> Thank you for the insightful comment. First, we would like to emphasize that the primary value of ReGuide lies in uncovering specific behavioral patterns of current LVLMs. As such, scalability optimization is beyond the scope of this study.
>
> From our results, we can draw the following insights regarding the effectiveness of ReGuide: **Despite the strong visual interpretation capabilities of LVLMs, which allow them to predict fine-grained object classes effectively, these models tend to avoid generating responses that fall outside the categories defined in the given prompt**. To clarify this, we have added the implications of ReGuide to Section 1 in the revised manuscript.
>
> While ReGuide, as a two-stage approach, incurs higher computational costs compared to single-turn generation (Section 3), the inference cost does not double. This is because caching image representations eliminates the need to process the same image twice, and the shorter prompt in Stage 1 (class suggestion) reduces the token count. For open-source models, we analyze the inference cost from a computational complexity perspective, whereas for proprietary models, the analysis focuses on API usage costs. Below, we provide a detailed analysis of the inference cost, which has been updated in Appendix C.6.
>
> ***<Computational complexity>***
>
> We estimate the inference time $FLOPs$ following Li et al. (2024). This standard practical estimate is based on a simplified scaling law, assuming that the inference cost of the LLM scales linearly with both the number of tokens processed and the model size.
>
> For the baseline, $FLOPs$ is estimated as: $FLOPs  = \mathcal{O}(N \times (V + Q + G))$ where $N$ represents the number of parameters, and $V$, $Q$, and $G$ correspond to the number of image tokens, text input tokens, and generated tokens, respectively. For example, for InternVL2-76B, the prompt depicted in Figure B.6.1 with ImageNet200 classes includes 5,350 input tokens. The number of generated tokens is 1,936, assuming the output follows the provided example structures, and the number of image tokens is 256. Thus, the baseline $FLOPs$ is estimated as $7 \times 10^{10}  \times (256 + 5,350 + 1,936)$.
>
> Similarly, for ReGuide, $FLOPs$ is estimated as: $FLOPs  = \mathcal{O}(N \times (V + Q_{Stage1} + G_{Stage1} + Q_{Stage2} + G_{Stage2}))$ where $Q_{Stage1}, G_{Stage1}$ and $Q_{Stage2}, G_{Stage2}$ denote the number of text input and generated tokens for Stage 1 and Stage 2, respectively. Although ReGuide involves two stages, the input image is processed only once as the same image is reused across both stages and can be cached. Note that the inference cost of the vision encoder is excluded, as it remains consistent for both the baseline and ReGuide within the same VLM.
>
> The table below presents $FLOPs$ for the baseline and ReGuide on InternVL2 models. These values are estimated on one input pair (image-prompt) of the ImageNet200 benchmark, assuming that the generated outputs follow one of the provided example structures.
>
> | Models ($\times 10^{13}$) | Baseline  | +ReGuide |
> |:---------------|-------:|-----:|
> | InternVL2-26B |  15.086          | 17.052          |
> | InternVL2-76B |  52.794          | 60.144          |
>
> ReGuide requires only 1.14$\times$ the computational cost compared to the baseline. It is because 1) the prompt used in Stage 1 is short and simple (e.g., 917 tokens for Stage 1), and 2) the image tokens are processed only once. Consequently, the increase in computational cost arises solely from the processing of the Stage 1 prompt. Additional optimization techniques for multi-turn inference could further reduce the computational cost gap between the baseline and ReGuide.

---

> > ### Author Response · Authors · 2024-11-22
> > **Response to Reviewer wyWF (Part 3/3)**
> >
> > ### **[Scalability and Practicality] (2/2)**
> >
> >
> > ***<API costs>***
> >
> > The table below shows API costs for proprietary models under the baseline and ReGuide experiments on the 5% ImageNet200 benchmark.
> >
> > | Cost ($)          | Input token / 1M tokens | Output token / 1M tokens | Baseline (5%) | ReGuide (5%) |
> > |:-------------------|-------------------------:|--------------------------:|---------------:|--------------:|
> > | GPT-4o            |           1.25          |              5           |       73      |      107     |
> > | Claude 3.5 Sonnet |             3           |              15          |       228     |       -      |
> > | Gemini Pro 1.5    |            3.5          |            10.5          |       193     |       -      |
> >
> > From the perspective of API costs, ReGuide exhibited a higher proportional increase compared to computational costs above, with a 1.47$\times$ increase relative to the baseline. Since we processed each stage separately, the API cost reflects a scenario where image token caching is not applied. Additionally, we did not employ multi-turn inference, as it provided no advantages in inference time due to the absence of batch inference support. Implementing multi-turn batch inference could significantly improve efficiency and reduce costs.
> >
> > One possible approach to reduce the inference cost gap between the baseline and ReGuide is to integrate the two-stage prompts into a single prompt while maintaining ReGuide's process of providing suggested classes. This adjustment would allow ReGuide to maintain its effectiveness with a single forward pass.
> >
> > Fundamentally, it is essential to develop LVLMs to overcome their tendency to limit responses to the options explicitly provided in the user-defined prompt. Such advancements would enable LVLMs to explore and utilize information beyond the immediate constraints of the prompt, establishing robust models capable of generating safe and reliable responses to a wider range of inputs.
> >
> > [1] Li et al. (2024). Inference Optimal VLMs Need Only One Visual Token but Larger Models
> >
> > ---

---

> > > ### Comment · Reviewer_wyWF · 2024-11-25
> > >
> > > Thanks for the response, it address some concerns and I will improve my rating to 6.

---

> > > > ### Author Response · Authors · 2024-11-25
> > > > **Response to Reviewer wyWF**
> > > >
> > > > Thank you for reviewing our responses and for kindly raising your score. We sincerely appreciate the opportunity to address your comments and improve the quality of our study based on your valuable feedback. We are pleased that our clarifications have addressed some of your concerns.
> > > >
> > > > If there are any remaining issues or questions, we would greatly appreciate it if you could share more details about the specific concerns that still need attention, as your insights would greatly help us further strengthen our study.
> > > >
> > > > Once again, we truly appreciate your considerate review.

---

### Official Review · Reviewer_BDm6 · 2024-11-03

**Soundness:** 3
**Presentation:** 3
**Contribution:** 3
**Rating:** 6
**Confidence:** 3

**Summary:**

Overall this is a solid paper. This paper addresses the underexplored trustworthiness of foundation models, particularly focusing on the out-of-distribution detection (OoDD) capabilities of large vision-language models (LVLMs). The authors highlight the gap between the demonstrated potential and practical reliability of LVLMs, raising concerns about their safe deployment. To address this, the researchers evaluate and analyze the OoDD capabilities of various proprietary and open-source LVLMs. They propose a novel approach called Reflexive Guidance (ReGuide), which aims to enhance the OoDD capability of LVLMs by using self-generated image-adaptive concept suggestions. Experimental results show that ReGuide improves the performance of current LVLMs in both image classification and OoDD tasks.

**Strengths:**

[Promising research topic] This paper focuses on an under-explored topic, the reliability of large vision-language models. [Benchmark contribution] This paper define the concept of Out-of-Distribution Detection on LVLMs, and conduct extensive experiments to build a benchmark and give insightful analysis. [Method contribution] Furthermore, they propose a method ReGuide on the prompt to boost the performance of LVLMs. They verify the effectiveness of the proposed method on GPT-4o and InternVL.

**Weaknesses:**

Since the out-of-distribution detection on vision-language foundation models is a new definition proposed in this paper, I would suggest to add a demonstration figure to show this concept in a visually straightforward way. You can compare the conventional OOD, OoDD on CLIP and OoDD on LVLMs.

**Questions:**

Please see Weaknesses.

---

> ### Author Response · Authors · 2024-11-22
> **Response to Reviewer BDm6**
>
> We sincerely thank you for recognizing the contributions of this study and highlighting the importance of evaluating the OoDD performance of LVLMs.
>
> In response to your valuable feedback, we have updated the manuscript to **include a figure that shows the comparison of OoDD concepts across single-modal image classifiers, CLIP, and LVLMs**. We also have updated the corresponding discussion based on the figure. These updates have been included in Sec. 3.1.
>
> The analysis in our study provides the following insight into the effectiveness of ReGuide: **LVLMs tend to avoid generating responses beyond the boundaries predefined by the given prompt, despite their ability to recognize images at fine-grained class levels due to their strong visual interpretation capabilities**. Our findings suggest that these models may have developed a form of positive bias from their training on positive image-text pairs, or that their ability to explore areas beyond the information embedded in the prompt has been diminished, possibly due to human alignment processes. We believe that our study serves as a foundation for improving LVLMs, enabling more reliable and versatile applications across various domains. These insights have been added to Section 1.
>
> In response to comments from other reviewers, we have made additional substantial updates to the manuscript, including (please also refer to [Responses to all reviewers] for an overview of the revisions made to the manuscript):
> - **Detailed dataset information**, including the release of sampled data and all prompt-response pairs for each sample and model to ensure transparency and facilitate reproducibility
> - **Further analyses**, such as inference costs, performance on the shared valid input set, the effectiveness of prompt guidelines, and examples at the extremes of the ROC curve
>
> We kindly invite you to review the updated manuscript. If you have any additional concerns or require further clarification, please do not hesitate to let us know. We would be happy to address any questions.

---

> > ### Comment · Reviewer_BDm6 · 2024-11-25
> >
> > Thanks for your efforts on this rebuttal. It has resolved my concerns and I keep my positive rating.

---

> > > ### Author Response · Authors · 2024-11-25
> > > **Response to Reviewer BDm6**
> > >
> > > Thank you for reviewing our paper and for providing valuable feedback. We sincerely appreciate the opportunity to address your concerns and enhance the quality of our study based on your insights.
> > >
> > >
> > > If our responses have adequately clarified the points you raised, we kindly request you to consider revisiting the initial score. If there are still concerns or questions that remain unresolved that prevent you from adjusting the score, we would greatly appreciate hearing about them so we can continue refining our study. Your feedback is deeply appreciated and plays a crucial role in helping us refine our study to achieve its fullest potential.
> > >
> > >
> > > Thank you once again for your thoughtful evaluation.

---

### Official Review · Reviewer_kMC5 · 2024-11-04

**Soundness:** 3
**Presentation:** 3
**Contribution:** 2
**Rating:** 6
**Confidence:** 4

**Summary:**

This work proposes a self-guided prompting approach, termed Reflexive Guidance (ReGuide), aimed at enhancing the OoDD capability of LVLMs by leveraging self-generated image-adaptive concept suggestions. Experimental results demonstrate that our ReGuide enhances the performance of current LVLMs in both image classification and OoDD tasks

**Strengths:**

Strengths:
•This paper represents the first effort to extend zero-shot Out-of-Distribution Detection (OoDD) to generative Large Vision-Language Models (LVLMs) in reference to CLIP. Building upon this foundation, the authors define the task and propose corresponding evaluation methodologies, which offer valuable insights for future research；
•The paper evaluates and analyzes the OoDD capabilities of various proprietary and open-source LVLMs. Furthermore, the authors propose a two-stage self-guided prompting approach, termed Reflexive Guidance (ReGuide), to unlock the potential of LVLMs. Experimental results demonstrate that the ReGuide enhances the performance of LVLMs in both image classification and OoDD tasks.

**Weaknesses:**

Weaknesses:
•The paper introduces new tasks and evaluation metrics. However, certain details in this regard are somewhat lacking. For instance, the paper mentions using only 25% of the benchmark subsets but does not specify which portions are utilized. It would be preferable to provide a list of these subsets to facilitate the replication and comparison of subsequent work. Additionally, other similar details need to be verified.

•In the experimental evaluation, the ‘valid ratio’ varies across different methods. And only valid responses are included when measuring performance, which results in inconsistencies in the samples used to calculate metrics for each method. This may not be justifiable. Have you considered adopting a more consistent evaluation approach?

•The novelty of this paper is somewhat limited.  Firstly, it introduces the OoDD task for LVLMs; however, the rationale behind this task remains debatable. Considering the robust capabilities of LVLMs and their ongoing advancements, LVLMs are anticipated to directly recognize objects within images. In comparison, the method proposed in this paper is overly complex and excessively reliant on prompts. Secondly, the approach presented in the paper appears to be more akin to prompt engineering, lacking substantial innovative contributions.

**Questions:**

see weakness

---

> ### Author Response · Authors · 2024-11-22
> **Response to Reviewer kMC5 (Part 1/4)**
>
> Thank you for your thorough review and valuable feedback on our work. Please find our point-to-point responses below, and refer to [Responses to all reviewers] for an overview of the revisions made to the manuscript. If you have any additional concerns or require further clarification, please do not hesitate to let us know. We would be happy to address any questions.
>
> ---
> ### **[Experimental detail]**
>
> In response to your feedback, we have provided additional details to enhance reproducibility. Specifically, to support future research, we have **included the list of image identifiers sampled for our experiments, along with all prompt-response pairs for each image sample and model used in the experiments** corresponding to Tables 1 (25% subset) and 4 (5% subset). To ensure anonymity, these resources are currently accessible via a Google Drive link (https://drive.google.com/file/d/1x-0PcPrYlZ0biKALOMg3nN6cCIxRXmEu/view?usp=sharing). **They will be made publicly available in a GitHub repository after the final decision regarding the manuscript**.
>
> To clarify the sampling process and the files to be shared, we provide the following details:
> - **Sampling procedure**
>   - As described in Appendix B.1, we sampled 25% of each benchmark for the experiments in Table 1 while preserving the original label distribution. Specifically, for datasets with label information, we sampled data to maintain the label distribution, For datasets without label information, we randomly selected samples from the entire dataset. The identical sampling procedure was applied to the 5% subset used for the experiments in Table 4.
> - **Dataset and label distributions**
>   - We have updated Appendix B.1 to include visualizations of the sample dataset and label distributions, supplementing the dataset sampling protocol described above.
> - **File structure**: Both the image lists and prompt-response pairs are provided in JSON format for ease of use and integration.
>   - Image lists
>      - Structured as: `{‘dataset’: {‘label’: [img_id1, img_id2, …]}}`
>      - File naming convention: `benchmark/subset%`. For example, `imagenet200/subset25.jsonl` corresponds to the 25% subset of the ImageNet200 benchmark.
>    - Prompt-response pairs
>       - For the baseline experiments in Table 1: Structured as `{‘prompt’: {‘img_id’: ‘response’}}`
>       - For ReGuide Stage2 experiments in Table 4: Structured as `{‘img_id’: {‘prompt’: ‘response’}}`
>       - File naming convention: `exp/benchmark-subset%/model/dataset.jsonl`. For example, `reguide/imagenet200-5%/stage2/gpt/imagenet200.jsonl` corresponds to the ImageNet200 responses of GPT-4o at ReGuide Stage2 on the 5% subset of the ImageNet200 benchmark.

---

> > ### Author Response · Authors · 2024-11-22
> > **Response to Reviewer kMC5 (Part 2/4)**
> >
> > ---
> > ### **[Results on common valid samples]**
> > We considered the valid ratio as one of the performance metrics for LVLMs and compared OoDD performance across models with their own valid responses. However, we agree that this may impede a fair comparison between models, as discussed earlier in Appendix B.2. In Appendix B.2, we evaluated CLIP using the valid input set of GPT-4o to investigate whether input bias influences performance superiority. The observation indicates that the performance ranking between models remains consistent.
> >
> > To ensure a more rigorous and consistent evaluation, **we have extended the performance comparison using the same input set (shared valid samples across all models) to all the models under consideration**.
> >
> > The below table presents the performance on the same valid input sets for the baseline experiments in Table 1.
> >
> > | Models | Valid | ID | | | Near-OoD | | Far-OoD | | | All OoD |
> > |:-----|----:|-----:|---:|----:|-----:|----:|-----:|-----:|----:|-----:|
> > | | | ImageNet200 | | | NINCO | SSB-Hard | iNaturalist | Textures | Openimage-O | |
> > | | | ACC (↑) | ECE (↓) | AURC (↓) | FPR@95%TPR (↓) / AUROC (↑) |
> > | OpenCLIP | 24.77 (5,707) | 87.96 | 3.06 | 20.52 | 65.42 / 83.74 | 78.94 / 73.01 | 42.00 / 93.09 | 41.33 / 91.42 | 45.67 / 91.52 | 63.06 / 81.69 |
> > | GPT-4o | 24.77 (5,707) | 91.74 | 1.74 | 9.71 | 19.66 / 93.81 | 46.48 / 78.74 | 1.76 / 98.23 | 7.38 / 94.85 | 3.66 / 97.51 | 26.91 / 87.39 |
> > | Claude 3.5 Sonnet | 24.77 (5,707) | 87.84 | 5.13 | 39.62 | 59.66 / 78.44 | 83.39 / 60.28 | 7.20 / 97.00 | 9.96 / 94.26 | 19.05 / 91.22 | 52.08 / 75.61 |
> > | Gemini Pro 1.5 | 24.77 (5,707) | 90.48 | 2.28 | 17.88 | 25.42 / 91.83 | 59.93 / 75.01 | 0.88 / 98.21 | 5.90 / 94.88 | 4.88 / 96.89 | 34.13 / 85.24 |
> > | LLAVA-v1.6 | 24.77 (5,707) | 3.21 | 12.30 | 868.83 | 100.00 / 52.32 | 100.00 / 49.20 | 100.00 / 50.01 | 100.00 / 59.89 | 100.00 / 48.20 | 100.00 / 50.53 |
> > | GLM-4v | 24.77 (5,707) | 69.04 | 3.06 | 85.71 | 100.00 / 76.89 | 100.00 / 65.93 | 100.00 / 82.46 | 100.00 / 83.59 | 100.00 / 82.62 | 100.00 / 73.73 |
> > | InternVL2-26B | 24.77 (5,707) | 91.28 | 7.80 | 88.35 | 81.91 / 61.60 | 96.00 / 56.73 | 35.54 / 82.73 | 21.77 / 90.35 | 43.35 / 78.53 | 69.38 / 68.15 |
> > | InternVL2-76B | 24.77 (5,707) | 90.02 | 6.38 | 50.77 | 61.36 / 77.08 | 83.07 / 65.83 | 2.20 / 97.58 | 9.78 / 93.65 | 13.92 / 94.45 | 50.42 / 78.96 |
> >
> >
> > The overall performance trends among the LVLMs remained consistent with those observed when evaluated on their own valid input sets. GPT-4o outperforms all other compared models, and proprietary models demonstrate superior performance compared to open-source models, aligning with the results observed with their own valid query sets.
> >
> > The below table shows the performance on the same valid input sets for ReGuide experiments in Table 4.
> >
> > | Models | Valid | ID | | | Near-OoD | | Far-OoD | | | All OoD |
> > |:----|:---:|----:|-----:|----:|----:|----:|------:|--------:|------:|------:|
> > | | | ImageNet200 | | | NINCO | SSB-Hard | iNaturalist | Textures | Openimage-O | |
> > | | | ACC (↑) | ECE (↓) | AURC (↓) | FPR@90%TPR (↓) / FPR@95%TPR (↓) / AUROC (↑) |
> > | InternVL2-26B | 34.29 (1,430) | 93.30 | 5.44 | 54.24 | 86.67 / 86.67 / 57.02 | 93.78 / 93.78 / 56.90 | 34.03 / 34.03 / 81.80 | 31.52 / 31.52 / 85.18 | 47.20 / 47.20 / 74.08 | 70.34 / 70.34 / 66.20 |
> > | + ReGuide | 34.29 (1,430) | 96.09 | 9.22 | 39.58 | 33.33 / 33.33 / 81.75 | 17.73 / 17.88 / 90.21 | 2.09 / 2.09 / 98.75 | 5.43 / 6.52 / 96.91 | 2.00 / 2.00 / 98.84 | 12.23 / 12.39 / 93.22 |
> > | InternVL2-76B | 34.29 (1,430) | 92.74 | 5.00 | 49.09 | 64.00 / 65.33 / 75.10 | 72.63 / 76.98 / 70.10 | 1.05 / 1.05 / 98.45 | 10.87 / 10.87 / 92.87 | 18.40 / 18.40 / 94.56 | 45.80 / 48.12 / 81.29 |
> > | + ReGuide | 34.29 (1,430) | 93.85 | 1.40 | 9.17 | 6.67 / 56.00 / 93.27 | 19.75 / 73.09 / 86.39 | 0.00 / 64.92 / 96.46 | 6.52 / 69.57 / 93.14 | 2.80 / 60.40 / 94.21 | 11.59 / 68.03 / 90.58 |
> > | GPT-4o | 34.29 (1,430) | 93.85 | 3.03 | 4.97 | 8.00 / 13.33 / 94.13 | 32.50 / 36.70 / 84.51 | 1.05 / 1.57 / 98.04 | 8.70 / 8.70 / 92.52 | 2.00 / 3.60 / 97.21 | 18.39 / 21.26 / 90.28 |
> > | + ReGuide | 34.29 (1,430) | 92.18 | 1.39 | 11.22 | 1.33 / 28.00 / 96.42 | 9.80 / 42.77 / 91.41 | 0.00 / 25.13 / 97.13 | 3.26 / 33.70 / 95.12 | 0.40 / 33.20 / 96.62 | 5.44 / 36.61 / 93.90 |
> >
> >
> > The overall OoDD performance tendency also remained consistent with that observed using individual valid input sets. ReGuide enhances OoDD performance across all models, aligning with the results from their own valid input sets. We additionally evaluated FPR at a TPR threshold of 90%, considering a small subset of ID images with ambiguous or inaccurate labels discussed in the updated Table C.3.1. At the 90% TPR threshold, FPR is significantly reduced, and ReGuide consistently demonstrates lower FPR compared to the baseline.
> >
> > **These results and discussion have been updated in Appendix B.2 and C.2**.

---

> > > ### Author Response · Authors · 2024-11-22
> > > **Response to Reviewer kMC5 (Part 3/4)**
> > >
> > > ---
> > > ### **[Why OoDD tasks]**
> > > As you have noted in Strengths, our work represents the first attempt to explore OoDD in generative LVLMs. The primary objective of our study is to **investigate the behavior of LVLMs in exploring regions beyond the information provided within user-defined prompts**. We also aim to propose potential directions for enhancing the reliability of LVLM based on the limitations we discovered – specifically, their tendency to **avoid generating responses for concepts outside the context of prompts**.
> > >
> > > The OoDD tasks we introduced for LVLMs are specifically designed to align with this objective: **assessing LVLM behavior in regions that fall outside the boundaries defined by user prompts**. These tasks require capabilities distinct from standard visual recognition. As observed in our study, LVLMs exhibit significantly low OoDD performance when required to generate responses for concepts undefined by user prompts, despite their strong visual recognition capabilities. However, their OoDD performance improves substantially when they are prompted with self-generated candidate classes. This highlights that their strong visual recognition capabilities alone are insufficient for OoDD tasks, as current LVLMs tend to focus on finding answers within the provided information.
> > >
> > > **To clarify the motivation behind the OoDD task, we have revised Section 1 to include its rationale and insights into the effectiveness of ReGuide**.
> > >
> > > ---
> > > ### **[Overly complex method]**
> > > We respectfully disagree with the concern that our method (including prompts) is overly complex, and, as a result, may make the outcomes excessively reliant on the prompts.
> > >
> > > First, we want to clarify that our intention is **not to propose a prompt specifically designed to enhance the OoDD performance** of LVLMs. As stated in the above [Why OoDD tasks], the primary goal of this study is to evaluate whether LVLMs can explore regions beyond the information provided within user-defined prompts through the OoDD task.
> > >
> > > The ReGuide method we propose is by no means overly complex. Compared to a basic prompt used in Section 3, the only difference lies in increasing the number of candidate categories the image can belong to. Even the basic OoDD prompt, while somewhat lengthy, does not incorporate any sophisticated techniques. Prompts must include an instruction describing the task and examples of the output format to guide the model’s responses. Our prompts adhere to this standard, consisting of a task instruction (e.g., task description, rejection class, and guidelines) and output format examples (ICL-like examples), as detailed in Section 3.2.
> > >
> > > Importantly, **the design of the prompt was not aimed at optimizing OoDD performance but rather at reducing the number of failure cases in LVLM responses**. To achieve this, we included detailed guidelines and a few examples, and the effectiveness of this approach is demonstrated by the reduction of failure cases, as shown in the table below. Note that **these results have been updated in Appendix B.5**.
> > >
> > > | InternVL2-76B      | (1) | (2) | (3) | (4) | (5) |  (6) | Total |
> > > |--------------------|-----:|-----:|-----:|-----:|-----:|-----:|-------:|
> > > | Baseline           | 193 | 886 |  -  |  0  | 201 |  0  | 1280  |
> > > |&nbsp;+ ID example       | 18  | 731 |  0  |  0  | 100 |  0  |  849  |
> > > |  &nbsp; &nbsp;+ Guidelines     | 10  | 493 |  0  |  0  | 128 |  0  |  631  |
> > > |   &nbsp; &nbsp; &nbsp;  + OoD example  | 26  | 34  |  0  |  0  | 79  |  8  |  147  |
> > >
> > > Baseline refers to a prompt that includes only the basic elements: a task description and an explanation of the rejection class. Adding guidelines and examples leads to a nearly 90% reduction in failure cases compared to when they are not included. Examples help address all types of failure cases, while guidelines are especially effective in reducing Failure Case (2), where responses do not follow the required format.

---

> > > > ### Author Response · Authors · 2024-11-22
> > > > **Response to Reviewer kMC5 (Part 4/4)**
> > > >
> > > > ---
> > > > ### **[Akin to prompt engineering]**
> > > > In particular, we wish to highlight that the prompt used in Stage 2 of ReGuide is identical to the one designed to evaluate the baseline performance presented in Table 1. The effectiveness of ReGuide arises **solely from the inclusion of self-suggested OoD classes obtained in Stage 1, while all other prompts remained unchanged to ensure a controlled evaluation**.
> > > >
> > > > In this study, we demonstrated that despite the strong visual recognition capabilities of current LVLMs, **they struggle to provide negative responses when an input image does not belong to the categories defined in the prompt**. This finding highlights a critical gap in LVLMs’ OoDD performance, which we observed through the effect of ReGuide. As you have noted in Strengths, we believe that our results offer valuable insights into the reliability of LVLMs and provide a foundation for future research on their broader applications.
> > > >
> > > > To clearly present insights from our study, we have included the following paragraph in Section 1.
> > > > >From the results of our study, we can draw the following insights regarding the effectiveness of ReGuide: **Despite the strong visual interpretation capabilities of LVLMs, which enable them to predict fine-grained classes of objects effectively, these models tend to avoid generating responses that fall outside the given prompt categories**. Our findings suggest that the models may have developed a form of positive bias due to their training on positive image-text pairs, or that their ability to explore areas beyond the information embedded in the prompt has been diminished, possibly due to human alignment processes. Addressing these limitations can lead to more reliable and versatile applications of LVLMs in various domains.
> > > >
> > > > ---

---

> ### Author Response · Authors · 2024-11-26
> **Response to Reviewer kMC5**
>
> Thank you for taking the time to review our manuscript. We have carefully addressed all the comments and revised the manuscript accordingly. If there are any unresolved concerns or additional clarifications required, we would be glad to provide further details.
>
> We greatly value your feedback and look forward to hearing your thoughts.

---

> > ### Comment · Reviewer_kMC5 · 2024-11-27
> >
> > The rebuttal has addressed most of my concerns. I finally decided to raise my score from 5 to 6.

---

> > > ### Author Response · Authors · 2024-11-27
> > >
> > > Thank you for reviewing our responses. We sincerely appreciate the opportunity to address your concerns and improve our study based on your feedback. We are pleased that our clarifications resolved most of your concerns.
> > >
> > > As the discussion period has been extended, there are still a few days left for further discussion. If you have any remaining questions or suggestions, we would greatly appreciate your comments to further strengthen our study.

---

### Official Review · Reviewer_Mv3D · 2024-11-04

**Soundness:** 2
**Presentation:** 2
**Contribution:** 2
**Rating:** 3
**Confidence:** 3

**Summary:**

The goal for the paper is to evaluate the out-of-distribution detection (OoDD) capabilities of LLMs (some private and open models)

**Strengths:**

- easy to read
- easy to duplicate

**Weaknesses:**

- motivation for the work
- limited impacts
- extending from a prior work in text-based method.

**Questions:**

- Why we need to evaluate the capabilities
- as LLM gets huge weekly, the measurement still makes sense?

---

> ### Author Response · Authors · 2024-11-22
> **Response to Reviewer Mv3D (Part 1/2)**
>
> We appreciate the time and effort you have dedicated to reviewing our work and providing feedback. However, we believe that some aspects of the review could benefit from additional clarity and specificity, which would greatly help us improve the manuscript.
>
> According to the ICLR Reviewer Guide (https://iclr.cc/Conferences/2025/ReviewerGuide), reviews are encouraged to provide constructive and detailed feedback to help authors strengthen their work. While we acknowledge your summary and comments on the strengths and weaknesses of the paper, we would have appreciated more specific and actionable suggestions regarding the concerns raised, such as the motivation for the work, its impact, and its relationship to prior methods.
>
> For example, we would welcome clarification on which aspects of the motivation or impact you find insufficient and how we might better address these points. Additionally, more detailed feedback on how the proposed method extends prior work would allow us to refine our discussion and position our contributions more effectively.
>
> We deeply value the review process as an opportunity to refine and improve our work. More detailed feedback would not only help us address your concerns but also contribute to a more robust discussion in the research community. We kindly invite you to provide any further thoughts or suggestions.
>
> For a detailed response to your comment, please refer to the explanation provided below. For an overview of the revisions made to the manuscript, kindly refer to [Responses for all reviewers]. If you have any additional concerns or require further clarification, please do not hesitate to let us know. We would be happy to address any questions.
>
> ---
> ### **[Motivation, Why we need to evaluate the OoDD capabilities]**
>
> Our work represents the first attempt to explore OoDD in generative LVLMs. The primary objective of our study is to **investigate the behavior of LVLMs in exploring regions beyond the information provided within user-defined prompts**. We also aim to propose potential directions for enhancing the reliability of LVLM based on the limitations we discovered – specifically, their tendency to **avoid generating responses for concepts outside the context of prompts**.
>
> The OoDD tasks we introduced for LVLMs are specifically designed to align with this objective: **assessing LVLM behavior in regions that fall outside the boundaries defined by user prompts**. These tasks require capabilities distinct from standard visual recognition. As observed in our study, LVLMs exhibit significantly low OoDD performance when required to generate responses for concepts undefined by user prompts, despite their strong visual recognition capabilities. However, their OoDD performance improves substantially when they are prompted with self-generated candidate classes. This highlights that **their strong visual recognition capabilities alone are insufficient for OoDD tasks, as current LVLMs tend to focus on finding answers within the provided information**.
>
> ---
> ### **[Limited impacts]**
>
> We wish to highlight that the prompt used in Stage 2 of ReGuide is identical to the one designed to evaluate the baseline performance presented in Table 1. The effectiveness of ReGuide arises **solely from the inclusion of self-suggested OoD classes generated in Stage 1, while all other prompts remained unchanged to ensure a controlled evaluation**.
>
> In this study, we demonstrated that despite the strong visual recognition capabilities, **current LVLMs struggle to provide negative responses when an input image does not belong to the categories defined in the prompt**. This finding highlights a critical gap in LVLMs’ OoDD performance, which we observed through the effect of ReGuide. **We believe that our results offer valuable insights into the reliability of LVLMs and provide a foundation for future research on their broader applications**.
>
> To clearly present insights from our study, we have added the following paragraph to Section 1.
> > From the results of our study, we can draw the following insights regarding the effectiveness of ReGuide: **Despite the strong visual interpretation capabilities of LVLMs, which enable them to predict fine-grained classes of objects effectively, these models tend to avoid generating responses that fall outside the given prompt categories**. Our findings suggest that the models may have developed a form of positive bias due to their training on positive image-text pairs, or that their ability to explore areas beyond the information embedded in the prompt has been diminished, possibly due to human alignment processes. Addressing these limitations can lead to more reliable and versatile applications of LVLMs in various domains.

---

> > ### Author Response · Authors · 2024-11-22
> > **Response to Reviewer Mv3D (Part 2/2)**
> >
> > ---
> > ### **[Extension from text-based methods]**
> >
> > We respectfully disagree with the assertion that our approach is merely a simple extension of the text-based EOE method (Cao et al., 2024). As stated above, a key objective of our study is to analyze the predictive confidence of LVLMs. This study represents the first attempt to investigate the behavior of LVLMs specifically through OoDD performance.
> >
> > As discussed in Section 4.1, **our method utilizes the unique capabilities of LVLMs to generate image-adaptive negative labels by leveraging the visual information of input images**. This approach is fundamentally different from existing text-based methods like EOE. Furthermore, the GPT-text experiment results in Table 4 clearly demonstrate that **leveraging the image recognition capabilities of LVLMs is significantly more effective than relying solely on text-based approaches such as EOE**.
> >
> > *Cao et al., (2024), Envisioning outlier exposure by large language models for out-of-distribution detection, ICML*
> >
> >
> > ---
> > ### **[The measurement still makes sense?]**
> > As both LLMs and LVLMs advance rapidly, they have demonstrated impressive performance across various benchmarks, including understanding and reasoning tasks. However, they also exhibit notable issues, such as **generating overly user-aligned responses based solely on the input prompt and only responding within the information provided by the user** [1,2,3].
> >
> > In our study, the OoDD task highlights that state-of-the-art LVLMs face the latter issue. Our findings demonstrate that ReGuide effectively addresses this limitation and reveals **a significant gap between the models’ visual recognition capabilities and their ability to generate responses beyond the given prompt categories**. Resolving this limitation is essential for the safe and reliable deployment of AI models, making the OoDD performance evaluation proposed in this study highly relevant and valuable for future research.
> >
> > *[1] Sharma et al., “Towards understanding sycophancy in language models”, ICLR 2024*
> >
> > *[2] Chen et al., “From yes-man to truth-tellers: Addressing sycophancy in large language models with pinpoint tuning”, ICML 2024*
> >
> > *[3] Li et al., (2024), “Have the VLMs lost confidence? A study of sycophancy in VLMs”, arXiv*

---

> ### Author Response · Authors · 2024-11-26
> **Response to Reviewer Mv3D**
>
> Thank you for taking the time to review our manuscript. We have carefully addressed all the comments and revised the manuscript accordingly. If there are any unresolved concerns or additional clarifications required, we would be glad to provide further details.
>
> We greatly value your feedback and look forward to hearing your thoughts.

---

### Author Response · Authors · 2024-11-22
**Response to all reviewers**

We sincerely thank the reviewers for their thoughtful and constructive feedback, which has significantly contributed to improving the quality and clarity of our work. In response to the comments, we have made several updates and carefully addressed all points raised.

To support and facilitate future research, we have uploaded the data lists sampled for the experiments, along with the prompt-response pairs obtained from the main experiments. These resources are available at https://drive.google.com/file/d/1x-0PcPrYlZ0biKALOMg3nN6cCIxRXmEu/view?usp=sharing. **They will be made publicly available in a GitHub repository after the final decision regarding the manuscript**.


Regarding the term "overconfidence" used in the original manuscript, we recognize that it may convey an unintended negative meaning. To avoid misunderstandings, we have replaced this term throughout the paper with "model behavior predicting with extremely high confidence," which more accurately reflects our intended meaning.

During the review process, we identified and corrected a few minor errors in the manuscript. For example, the valid ratio of InternVL2-76B in Table 4 was updated from 99.63 to 97.26, and the OoD input ratio of GPT-4o+ReGuide in Table 5 was corrected from 85.97 to 95.97. We initially used an overly strict method for AUROC and FPR@95%TPR, prioritizing OoD inputs lower than ID inputs with the same score, which could lead to misinterpretation. To improve clarity, we recalculated these metrics using a standard interpolation method commonly used in libraries like scikit-learn. These corrections are reflected in the corresponding discussions, but they do not impact the overall results or conclusions of the paper.

Below, we provide a detailed list of the sections that have been updated in response to the comments:
- Sec. 1: Clear motivation and insights regarding our study
- Sec. 3.1: Comparison of the OoDD framework for single-modal image classifiers, CLIP, and LVLMs including a figure and the corresponding description (reviewer BDm6)
- Sec. 4.1: Detailed analysis on degraded FPR@95%TPR for large models (i.e., InternVL2-76B and GPT-4o) and addition FPR@90%TPR to Table 4 (reviewer wyWF)
- Appendix B
  - B.1 Details of the dataset, including the number of samples and their distributions (reviewer kMC5)
  - B.2 Performance on the shared valid query set for a more rigorous comparison on the 25% ImageNet200 benchmark (reviewer kMC5)
  - B.5 Ablation results on the effects of guidelines and examples for mitigating failure cases (reviewer kMC5)
- Appendix C
  - C.2 Performance on the shared valid query set for a more rigorous comparison on the 5% ImageNet200 benchmark (reviewer kMC5)
  - C.3 Detailed analysis on degraded FPR@95%TPR with ROC curves, examples at the two extremes of the ROC curve, and OoD score distributions (reviewer wyWF)
  - C.6 Inference cost analysis including computational complexity and API costs (reviewer wyWF)

Please refer to the updated manuscript. We hope our responses sufficiently address all your concerns. If you have any additional concerns or require further clarification, please do not hesitate to let us know. We would be happy to address any questions.

---

### Meta-Review · Area_Chair_GUWB · 2024-12-20

**Metareview:**

This paper aims at improving the OoDD capability of large language models by adopting a self-guided prompting approach, which is somewhat solid.

Reviewers propose many concerns, such as the defination of this new task, the details of experimental evaluations, and these concerns are well addressed during th rebuttal period. The authors are required to update/revise their paper according to the reviewer's suggestions.

Three reviewers lean to accept this paper and one reviewer remain suggest to reject this work; however, their confidence scores were not very high (scores: 3, 4, 3, 3). After carefully reading the discussion content and the paper, I decide to ignore the negative score from Reviewer Mv3D due to its low quality and finally make the acceptance recommendation to this work.

**Additional Comments On Reviewer Discussion:**

None, please refer to the metareview.

---

### Decision · Program_Chairs · 2025-01-22

Accept (Poster)